# Structural features of the streetscape of Macau across four different spatial scales based on historical maps

Youping Teng[1,2], Shuai Yang[1,2]*, Yue Huang[3]

1 Department of College for Creative Studies, Zhejiang University City College, Hangzhou, Zhejiang, China, 2 College of Computer Science and Technology, Zhejiang University, Hangzhou, Zhejiang, China, 3 Faculty of Innovation and Design, City University of Macau, Macau, China

* samyang@zju.edu.cn

**Data Availability Statement:** Data are available on figshare: Shuai Yang (2021): Structural Features of Streetscape in Macau at Four Different Spatial Scales, Based on Historical Maps, https://doi.org/10.6084/m9.figshare.16714318.v1.

## Abstract

In the analysis of complex historical layering, studies on how to avoid theoretical analysis and use quantitative methods of display and analysis are scarce. Therefore, we used space syntax to fill this gap in historical layering analysis. We established a spatial digital model by combining the urban historical landscape theory with the space syntax analysis method. Then, we quantitatively analysed the streetscapes in the four historical periods of Macau and the value-related development of the city's economy, society, and culture. To this end, we used the theory of urban historical landscape to interpret the streetscape of Macau. We reviewed urban development under different spatial scales, which represent different states of historical layering. Changing ideological trends of construction have induced changes in the city, which have led to changes in the city style. The analysis considers the dimensions of space and time, and its results can guide the continued benign growth of the urban landscape and solve protection problems in practice. Meanwhile, the results of this work also indicate that the unique streetscape of Macau bred by the development of the city does not affect the newly constructed roads. The newly reclaimed areas and the streetscape of the new city are on the verge of homogeneity and cannot reflect the unique regional characteristics of Macau. Therefore, we used the historical map of Macau as a carrier, used space syntax to analyse the structure of Macau's streetscape, and explored the apparent characteristics and value associations carried by the streetscape of Macau under different historical slices. Our results can help retrieve the value of Macau's historical streetscape and devise a targeted protection strategy that can help pass on the historical streetscape of Macau to posterity.

## Introduction

The urban historical landscape emphasises starting from the larger-scale urban background and its geographical environment, and it attaches importance to continuation of the context and spirit of a place. Although the term 'layering' has roots in geography, it is used to describe

**Funding:** This was funded by China Postdoctoral Science Foundation, grant number 2021M692781, and by Projects of the National Social Science Foundation of China, grant number 16BSH036. The funders had no role in study design, data collection and analysis, decision to publish, or preparation of the manuscript.

**Competing interests:** The authors have declared that no competing interests exist.

diverse responses of different cultures in the same time dimension. 'Layer' refers to the differentiation of time, that is, the juxtaposition of different cultures across different historical periods. 'Accumulation' refers to the interrelationships among these cultures outside time differentiation, such as overlap and conflict with each other [1]. However, the reflection of a single building on history and culture can highlight only one characteristic difference. At the block and city levels, the superposition of historical relics from different periods and belonging to different cultures can be observed in the same space [2]. Whether one considers the perspective of time differentiation or cultural overlap, one must maintain a long-term and continuous perspective [3]. The United Nations Educational, Scientific and Cultural Organization (UNESCO) published an instructional collection of essays in 2017 detailing the importance of historical layering in the study of historic urban landscapes and conducted a detailed case study of historical layering in Cuenca, Ecuador [4]. Macau and Cuenca exhibit many similarities since their initial rise periods. Both the cities used to be colonies, and under the continuous action of time and space, continuously historical layering is occurring in both the cities. In these cities, spatial parts are maintained; some parts are replaced or declined. Therefore, the aforementioned UNESCO study provides an important theoretical basis for studying historical layering in Macau.

The law of urban evolution in the study of historical layering is based on the wisdom of scholars and provides a clear context for urban development. The experience and lessons contained in the streetscape layering study provide valuable reference for the development of modern cities. The wisdom of our ancestors, such as cultural traditions, construction technology, morality and etiquette, is worth inheriting and carrying forward. Moreover, it also provides a few positive countermeasures for the future development of the city. The study of 'layering' in the Vienna Memorandum signed at the International Conference on 'World Heritage and Contemporary Architecture' held in Vienna in May 2005. Economic development, social function evolution, and political intervention, among other phenomena, have led to limited unnatural development to a certain extent in the development of urban landscapes [5]. The concept of layering has been emphasised many times in the proposal. Initially, layering was applied in geography as a manifestation of the historical accumulation of geology, and subsequently, it was used to describe the accumulation of culture [6]. In geography, layering emphasises the accumulation and coverage of geology, and time is estimated through different strata. However, from the perspective of urban historical landscapes, layering serves as an objective reflection of multi-existent cultural features in different time dimensions. When expounding the concept of historic urban landscapes, the concepts of archaeology, history, aesthetics, and architecture are used. The Vienna Memorandum provides a theoretical basis for the subsequent concept of 'layering' [7]. According to the concept of urban historical landscape, the development of a city is the accumulation of cultural and natural values over different periods. In this sense, the historical landscape of a city is the result of historical 'layering'. I believe that the concept of historical layering can be summarised as the accumulation of multiple historical values of a city since its formation, such as culture and traditions, under the influences of economy, society, culture, and other human factors [4]. To study the historical layering of Macau, we must possess a thorough understanding of urban physiognomy to ensure that the heritage value penetrates into all aspects of the historical heritage protection. It is well recognised that Macau's urban physiognomy has grown out of a lengthy confluence between the traditions of Luso-Mediterranean cities and native Chinese settlements. Since the formative years of Macau, the fortified Christian city, ordered by strategic placements of civic and religious institutions, was surrounded by the Chinese city with the bazaar at its heart. Together with the hilly peninsular topography and specific urban functionalities, a unique spatial pattern of squares (largos), public courtyards, and narrow contour-hugging streets and

alleys (ruelas, travessas) evolved to house its civic life and culture [8]. From the urban morphology perspective, the urban form of Macau shares a few similarities with those of many cities. Macau, for example, is facing serious challenges due to traffic congestion and urban renewal, which in turn hinder social, economic, and environmental development. The impact of Macau's archipelagic status on its development remains unknown: increasing land reclamation has caused the fragmentation of space-time cities. Based on data from the literature, this paper expounds the historical development of Macau and analyses its urban fragmentation and the impact of this fragmentation on the urban environment of Macau [9].

In the literature that has applied the concept of urban historical streetscapes, OERS.R.V and Bandarin F. presented a new interpretation of the concept of urban historical landscape from the perspective of methodology and applied it to numerous practical cases [10]. Rodwell D. introduced the revitalisation process of the central city of Liverpool, England, and discussed the important role of urban historical landscape [11]. De Rosa F. and Di Palma M used the preservation of Naples, Italy, as an example to discuss the relationship between the preservation of regional characteristics of historical landscapes and the development trend of globalisation and how to balance the two [12]. Gordon A and Simondson D combined the analysis of urban landscape evolution mechanism with the protection of urban regional features and proposed a land use planning method that integrated urban landscape protection [13]. In terms of the methodological studies on urban historical streetscapes, in addition to space syntax, Lynch K, Cullen G., and other scholars adopted the method of base analysis and made outstanding contributions to SITE research and analysis. In his book Site Planning, Lynch K. proposed a set of site analysis methods that consider the social, cultural, psychological, natural, and material elements of a site, and this work has significantly influenced modern urban design [14]. Cognltive Map, an urban spatial analysis technology borrowed from cognitive psychology, is another effective method for studying urban historical streetscapes. The concrete process uses the sociological survey method for reference, which is effective for controlling an urban landscape and place images. The specific approach involves investigating residents' psychological feelings and impressions of a city through inquiry or written methods, and subsequently, allows designers to analyse the results of the survey and to generate maps, or more directly, encourage the residents to draw sketches of the spatial structure of the city [15]. In addition, drawing 'landscape narrative historical map' is another method that expresses the changes in the landscape of a certain region by using maps. This method was applied to Shanghai as an example to elucidate the process of generating historical and cultural maps of the city [16, 17]. In this paper, in the process of studying the streetscape of Macau, to better explain the layering changes of different periods, we use the space syntax method.

In the past, owing to the limitations of technical means and research methods, it was difficult to conduct deep quantitative analyses. However, the development of spatial analysis tools in recent years has created new opportunities for quantitative research [18]. Space syntax was pioneered in the 1970s by Professor Hillier from the University of London [19]. Space syntax is based on the principle of graph theory for modelling buildings and urban spaces and analysing the relationship between spatial morphological features and their social functions [20]. The space syntax theory provides us with new ways to know and understand space, in addition to helping us understand 'how space itself affects culture and society' [21]. As a theory, space syntax is different from classical urban form; it is a new language to describe architectural and urban spatial patterns. The fundamental idea of space syntax is to divide space and analyse its complex relationship [22]. The space referred to in space syntax is not only the object measured by mathematical methods (described in Euclidean geometry) but also the relationship among spaces such as topology, geometry and actual distance. The introduction of space syntax theory spurred quantitative data analysis. Space syntax is used to analyse street and lane

networks in GIS environments. In this manner, the close interaction between spatial forms and human behaviours can be revealed scientifically, and the relative importance of each space in the entire spatial system can be studied easily by analysing data and images. It is a more intuitive and quantitative research method, and it reflects the form and development of streetscapes in a particular historical period through various syntactic indicators. In this study, we use the space syntax theory to study the structural layering of streets and lanes in Macau. To this end, we perform spatial syntax analysis of the historical map data of 1780, 1889, 1950, and 2018. This analysis of the characteristics of the street network across different historical sections is the basis of the authors' historical layering analysis. We combine the ancient map of Macau with space syntax to explore the quantitative research methods of layering across different historical periods. The concept and method of urban historical landscape provides us a new perspective for understanding and recognising the changes in the layering of the entire city. The main purpose of this study is to arouse people's cognition of urban historical and cultural heritage.

## Materials and methods

### Study area

In its long history, China has experienced several heydays of cultural exchanges with the West. The exchange of personnel between China and the West, cultural transmission of religions by missionaries, and mutual transplantation and donation of products have all positively affected the exchange of literature, art, economy, and ideas between China and the West [23]. In the 16th century, China was ruled by the Qing Dynasty, which isolated the country from the outside world. From 1553, Portugal gradually began to colonise Macau. For many years thereafter, Macau, as an important foreign trade port, served as a window for foreigners to enter China. The Portuguese paved roads and constructed residential houses, churches, and batteries in Macau, in addition to introducing European ideas and ways of life to Macau. Since then, people from Portugal, Spain, Italy, Britain, Sweden, and many other countries conducted trade in Macau, and Macau quickly emerged as a global city and international port. Objectively, Macau is one of the earliest cities in China to have come into contact with Western culture [24].

During the 400 years of Portuguese rule in Macau, the coexistence and co-integration of diverse cultures has been a typical urban feature of Macau. In terms of culture, Portuguese cultural thoughts have constantly been integrated with the Oriental culture of Macau, and the peaceful coexistence of the two cultures has created a harmonious state of co-prosperity and co-dependence between them. This state is especially reflected in the mutual respect and coexistence of Eastern and Western religious and ethnic cultures. Macau advocates freedom of religious belief, and its religious culture can be considered all-embracing. Buddhism, Taoism, Matsu belief, Islam, Catholicism, Christianity, Hinduism, and other sects coexist respectfully and peacefully in Macau and leave space for each other, which is a manifestation of the multifaceted religious culture of Macau. In addition, for historical reasons, the Chinese culture, native Portuguese culture, and Portuguese culture have converged in Macau, and Macau has emerged as a region where multi-ethnic cultures blend. In terms of architectural art, the Portuguese government has constructed many buildings with Portuguese characteristics in Macau, and the unique cultural characteristics of Macau have really penetrated into the urban buildings of Macau. Portuguese-style churches were built at many locations in Macau, and Portuguese-style gravel paved roads are common across the city. Interestingly, these buildings are not in conflict with the local buildings of Macau, nor are there any visual dissonances. This can be ascribed to the tacit understanding and mutual tolerance cultivated by the Portuguese and Macau people over the years. Among the architectural wonders of Macau, the historical

buildings of Macau are the most valuable. Due to the premature artificial maintenance and protection of the historical buildings in Macau, strict mechanisms and modern technology are used to repair or reproduce of the original features of the building and to reproduce their historical features. The preservation of historical buildings is important to witness the harmonious coexistence between Portuguese people and Chinese people and as a precious treasure trove for studies on world cultures.

The urban streetscape of Macau is an important component of the regional characteristics of Macau. The streets depict the continuation of the historical memory of Macau and have irreplaceable historical value. Apart from the streets themselves, the buildings on both sides and the ground, too, are parts of the street space. The most representative street is the straight street of Macau. This special urban fabric confirms the characteristics of Chinese and western cultures in Macau and represents the historical activities, behaviours, and living habits of Macau people. It is a comprehensive space with the mutual integration of functions and materials. Under the accumulation of >400 years of history, Macau, where the Chinese and Western cultures meet, has formed an indispensable part of Macau's urban landscape. After years of layering, the streetscape of the old city retained its inherent value, attracting tourists from all over the world to stop and stay. From the perspective of urban development, the streetscape in the urban historical landscape represents the precious heritage of the city.

## Data acquisition and processing

**Layering course of streetscape in Macau.** The urban streetscape develops along with the development of a city's historical landscape. The urban development context of Macau is divided into four stages: initial stage (1557–1586), growth stage (1586–1840), heyday stage (1840–1974), and decline and regeneration stage (1974–present). Moreover, the urban streetscape follows and inherits the development context of the city, and it is synchronously divided into the aforementioned four stages. This study focuses on these four historical stages and completely utilises the visual advantages of space syntax to extract and analyse the structural and value associations of the streetscape of Macau [25, 26].

We vectorised the official historical maps for the years 1780, 1889, 1950, and 2018. These maps were used as the base maps for layering analysis and self-drawing. The base maps for the years 1780, 1889, and 1950 were obtained from the Archives of Macau operated by the Cultural Affairs Bureau of the Macau S.A.R. Government. The base map for the year 2018 was obtained from USGS National Map Viewer [27]. This paper is based on the study of historical maps of Macau across various periods. Because the historical maps were drawn manually, we can draw the axis map on the basis of the historical maps by applying some corrections and through other interventions; however, we cannot avoid the possibility of small deviations from reality. Firstly, we vectorised each map and used AutoCAD (2018) to obtain the street and lane structure drawings of the four periods. Then, according to the drawings, through ArcGIS (10.2) and space syntax, the data of integration and travel degree, respectively, were derived for the four periods. The model results were assigned to the geographic coordinate system WGS1984, and spatial correction was performed. Secondly, we imported the calibrated data into Depthmap software to start calculation of the spatial syntax, set the local variable radius to 500 m, and set the global variable radius to 'n'. Finally, we converted the calculated result into a shape file, which we subsequently imported into ArcGIS for mapping and spatial analysis.

## Data analysis

There are many models of space syntax, and the present study mainly uses the space syntax line segment model [28]. The line segment model is a special case of the axis model. The line

segment model uses spatial fragments for modelling. The line segment model based on the road centreline is more conducive to analysing large-scale/urban-scale roads than the axis, convex, and VGA models. Moreover, this excellent model considers the correlation between road network deflection and human movement. The main difference between the theoretical bases of the line segment and axis models is the assumption that travel in space follows the rule of 'the smallest deflection angle' in the former model. Therefore, the model assigns weights to the angles between road segments in the calculation of road accessibility. Because the modelling cost of the line segment model is lower than that of the axis model, and the model is intuitive and has small errors, it has emerged as the mainstream model of space syntax in urban space research and planning [29–31]

The main syntactic variables used in this article are as follows:

1. Angle integration. In the context of the line segment model, the methods for calculating many spatial syntax parameters have been modified. Angle selectivity in the context of the line segment model is no longer standardised by the 'diamond shape' but is directly expressed as the quotient of the number of nodes within the search radius and average angle depth. Under the line segment model, the angle is considered the incident angle between the road centreline segments. Generally, the included angle is between 0 and 180 degrees, and a value of 0–2 is assigned as a weight. The angle distance is computed as the weighted sum of the incident angles of the road segments covered by a path in the same direction. Similar to axis integration, line segment integration reflects the degree of agglomeration or dispersion between a certain space and other spaces in the network. The higher is the integration, the greater is the spatial agglomeration, and the stronger is the topological accessibility of the space. Integration is an important parameter for measuring the permeability and reachability of space.

The formula for calculating integration is as follows [32]:

$$Integration = \frac{n * n}{\sum_{i=1}^{n} d\theta(x, i)} \tag{1}$$

In the formula, integration represents the integration of line segments, n represents the total number of nodes within the search radius, and $d\theta(x, i)$ represents the angular topological distance between spaces x and i.

2. Angle choice. Angle choice is an important parameter in space syntax. However, owing to standardisation issues, angle choice is rarely used in conjunction with the traditional axis model. In case of the line segment model, Tao et al. solved the standardisation problem of angle choice. Angle choice reflects the probability of a certain space node being passed. The higher is the angle choice, the greater is the probability that a space is passed through. Angle choice can often be used to identify roads with high traffic unblock capacity. The formula for determining angle choice is as follows [33]:

$$NACH = \frac{log\left(\sum_{i=1}^{n}\sum_{i=1}^{n}\sigma(i,x,j) \Big/ (n-1)(n-2)\right)}{log(\sum_{i=1}^{n} d\theta(x,i) + 3)} (i \neq x \neq j) \tag{2}$$

In the formula, $\sigma(i, x, j)$ denotes the number of paths passing through space x while starting from space i and ending in space j, and $d\theta(x, i)$ denotes the angular topological distance between spaces x and i. Hillier's empirical evidence proves that angle choice is independent of the total number of nodes in the road network but is highly correlated with the total depth of the space.

3. Intelligibility. It is difficult or even impossible for us to experience a large-scale space in situ at once. To achieve this goal, we must experience each small-scale space in motion to

understand the overall space. Intelligibility characterises the difficulty experienced by a user in gradually establishing a picture of the entire space system by observing local spaces.

Intelligibility is a common phenomenon when recognising large-scale spatial structures such as buildings and cities. Space syntax judges the intelligibility of a spatial layout by establishing the relationship between local space and overall space. According to the definition of intelligibility, if the information (connected value) directly obtained by a user from a space can help them achieve a good understanding of the overall space (integration), then the spatial system is understandable or is easy to understand. Intelligibility emphasises the overall situation of the system; this, it is a global spatial variable.

Alternatively, we defined intelligibility as the correlation between integration and connected values. Correlation is used to describe the degree of correlation between local control and overall association. It reflects the ease with which a subject can distinguish a local space pattern from the overall space pattern. Local space and global space must exist, and there is a certain degree of correlation between them. The stronger is this correlation, the more consistent is the local perceptible space with the overall invisible space, indicating that the overall spatial form is easier to understand [34]. More precisely, intelligibility is the degree of linear correlation between integration and the connected value of all small-scale spaces in a space system. The stronger is the relevance, the higher is the intelligibility of a space, and the easier it is for users to infer the overall space layout from the spatial information directly available to them.

## Results

### Summary of explicit characteristic layering rule of the streetscape in Macau

We performed a detailed analysis of the layering evolution process of the road network of the streetscape of Macau for the four historical periods. During the developmental years of Macau, under the time periods of the four historical cycles, representative roads and spatial nodes were selected to interpret the historical contexts of the dynamic evolution of the streetscape by interpreting the historical maps. Moreover, we analysed the layering of the road network structure in the road landscape of the Macau Peninsula to obtain the changes and associations between layers and the spatial changes and associations beyond their representation.

### Streetscape of Macau in the initial stage (1557–1586)

A large amount of wealth has been transported from Macau to various places for trading, and it has emerged as a prosperous port city for maritime trade. Therefore, the citizens of Macau are richer than those of other cities. Overseas trade been well developed, and the transactions among Chinese businessmen in the city are highly sophisticated. The Farangi Biography in the History of the Ming Dynasty states that Fujian and Guangdong merchants rushed to Macao to trade [35]. The historical documents show that Macau was a transit point for goods, and the trade was also prosperous with the Chinese in the city, that is, the 'Yi Market in Macau' appeared [36, 37].

### Streetscape of Macau in the growth stage (1586–1840)

According to the historical map of Macau (Fig 1), in the growth stage, the urban spatial structure of Macau inherited the spatial structure of the city under the rule of the Portuguese. The street and lane landscapes are the typical representatives. Its main manifestation is a straight street with an irregular linear structure, which is connected to churches and the characteristic

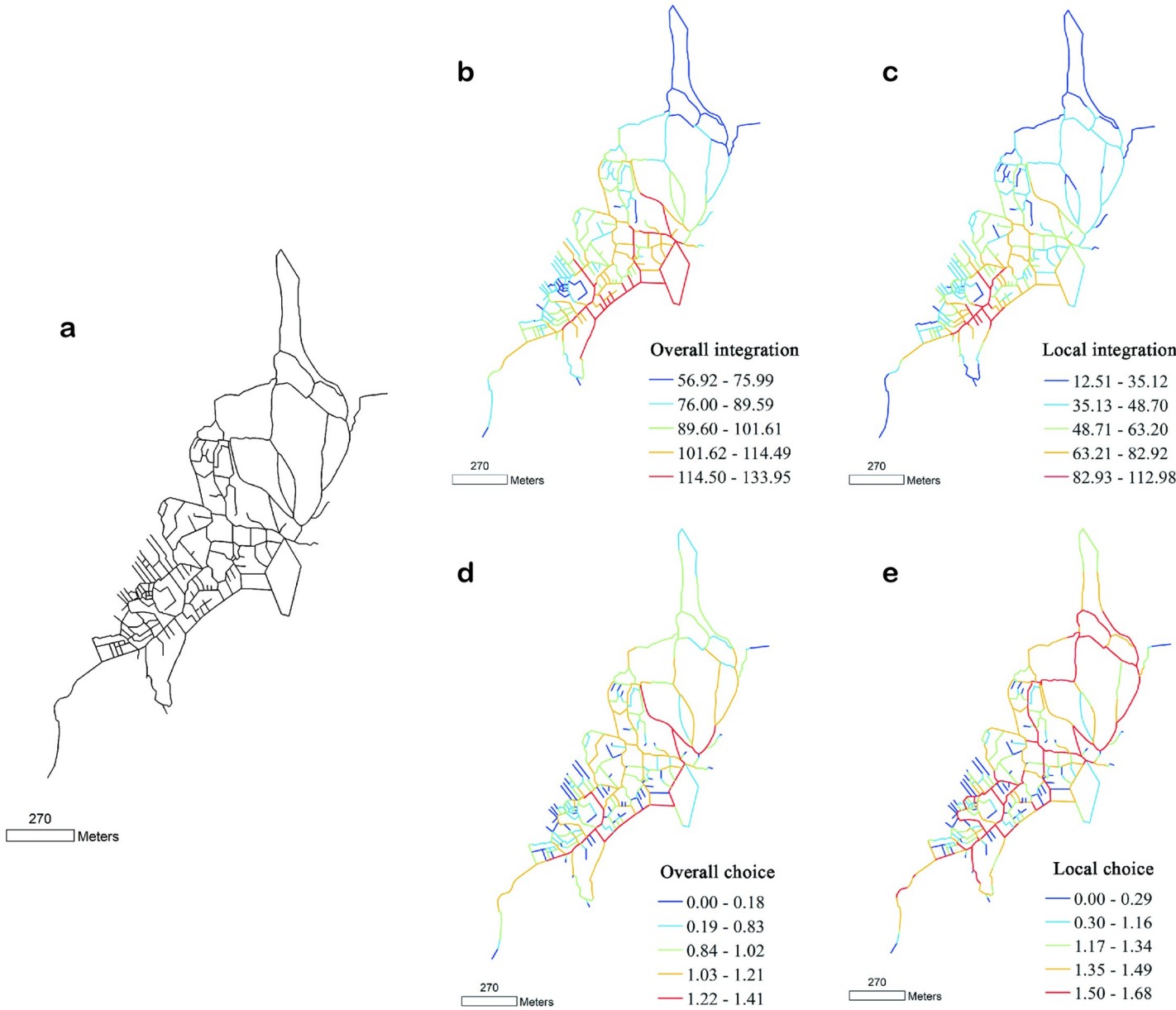

**Fig 1. (a) Structure, (b, c) integration, and (d, e) angle choice of Macau streets and lanes, drawn according to the 1780 base map.** Global variable radius = n; local variable radius = 500 m.

front land of Macau. This form is a typical symbol of the Portuguese cities from the Middle Ages, namely the Renaissance period.

According to the organisational structure of the road network of Macau's streetscape in 1780, the core area of the spatial structure shows a bias towards the characteristics of the Pucheng District of Nanwan. The Portuguese city is the core, and the westward urbanisation is not obvious. During this period, the Portuguese built many church buildings within the city walls, which is the main building type for the Portuguese. At the same time, they built factories, hospitals, schools, and other residential facilities, as well as large numbers of public buildings. The urban structure with 'straight streets' as the skeleton has always been a characteristic of Macau, which makes it more compact and complete.

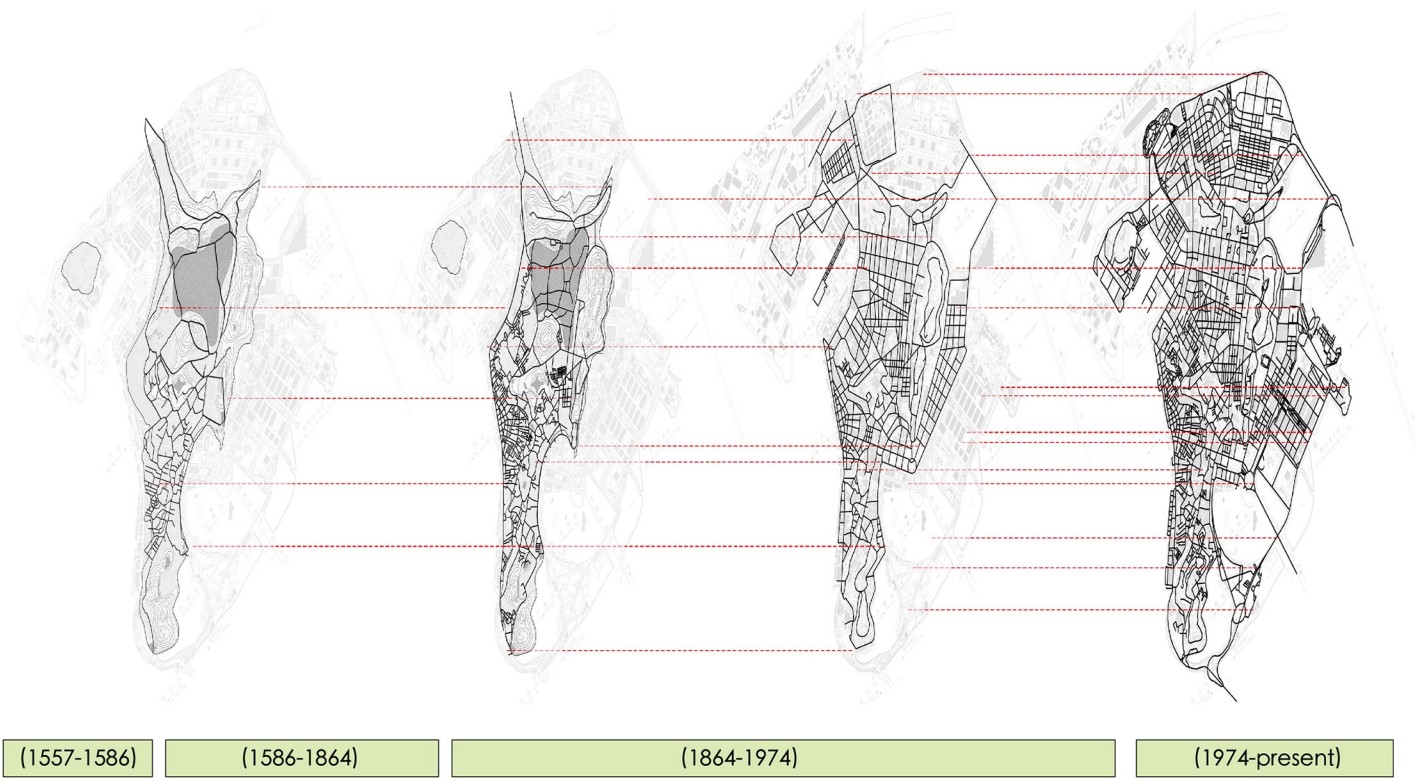

| (1557-1586) | (1586-1864) | (1864-1974) | (1974-present) |

**Fig 2. Road layering of the Macau Peninsula, based on drawings from maps of four different time periods.**

In the early stages, the west side of the city was a port for maritime trade. After years of sediment accumulation and continuous renovation projects by the Chinese, it emerged as an urban area dominated by the Chinese. The layout of streets and lanes in this area builds on the Chinese tradition. The entire block has a rectangular layout. The streets are developed parallel to each other and extend towards the port. The historical maps indicate that this area is the current inner port area, where the Chinese population is dominant and is engaged in business, and it is attached to the Pucheng District [38].

According to the space syntax, the higher is the integration, the greater is the spatial agglomeration and the stronger is the topological accessibility of the space. Integration is an important parameter to measure the permeability and reachability of a space. In Fig 2, the higher is the integration, the greater is the urban spatial agglomeration of Macau and the stronger is the accessibility of its streets. Moreover, the area with the highest overall integration is located at the centre of the business district in Macau (Fig 2). The specific area is the southeast coastal commercial area in the Macau Peninsula, which is distributed along the straight streets of Macau. The middle part exhibits a high level of high overall integration, and the north and west parts exhibit lower levels of overall integration. These results indicate that urban streets are mostly distributed around the straight streets, and the remaining areas have poor accessibility.

Overall, in this period, the road network of Macau did not have a continuous structure. Especially in terms of traffic-carrying capacity, the traffic-carrying capacity of each road was not uniform. Therefore, at that time, the road network of Macau had considerable 'native' characteristics, and the irregular shape of the road network reflected that the topography of Macau strongly influenced the road network.

## Streetscape of Macau in the heyday stage (1840–1974)

The period from 1840 to 1974 was the heyday of urban development in Macau. In 1840, the Portuguese–Macau government expanded towards the villages north of the city wall. From 1848 to 1866, a 'two longitude and two latitude' road network was built in the northern part of the city. After 1866, the new road network was expanded into a 'two longitude and four latitude' pattern. During 1889–1912, a 'six longitude and nine latitude' road pattern was established, and the urban expansion of the northern part of Macau was completed [39].

According to Fig 3, in 1889, the core area of overall integration was located in the newly reclaimed area on the west coast of the Macau Peninsula. Due to geographical and environmental reasons, reclamation activities were first conducted in Beiwan and Shallow Bay, which included straightening of the coastline from Sidakou to Xianyu Square, opening of the Hebian New Street, and initiation of a reclamation project outside the Riverside New Street. This project was completed in 1881. The pattern of newly built streets was different from the streets that were developed organically in the slightly messy inner harbour. The new streets were grid-shaped and arranged lengthwise along the coastline in the traditional 'comb layout'.

Based on an analysis of the streetscape structure in the Macau Peninsula in 1950 (Fig 4), we concluded that with the urban renewal and the emerging reclamation project, the development pattern of the new streets with a three-grid outline is clearly visible on the map, and Macau entered a new era at this time [40].

An analysis of the characteristics of the overall choice of road network in Macau in 1950 (Fig 4) revealed that after the continuous reclamation project in northern Macau, the seawall road was connected to Chizhou Island, and this road connecting the old city with the newly reclaimed area had a thin chopstick-like appearance. Thus, a network characterised by high overall choice value was formed in the city, which constituted the main traffic axis outside the old city with new traffic arteries such as the Mei Fujiang Road. In the old Macau city, the traditional 'Straight Street', which was rebuilt as a part of the new road network, emerged as an important city commute route again. In 1950, the areas with higher local choice in the road network of Macau were the road network around the east and west Wangyangshan Moutain, Xinma Road, and Shuikengwei Street. This indicates that the renewal of historical blocks yielded initial success, and the core of the city was still concentrated in the old city area. The concept of protecting the old city laid the foundation for the future urban renewal and protection, which continues to have an impact to this day.

## Streetscape of Macau in the decline and regeneration stage (1974–present)

In this stage, the sovereignty issue of Macau has been gradually resolved. In terms of urban development, various major transportation facilities have been constructed to expand the urban development space. According to the 2018 Macau Peninsula Street Structure Map (Fig 4), the spatial structure of the city is getting closer and closer, and the density of the street network is increasing.

An analysis of the organisational structure of Macau's road network in 2018 (Fig 4) revealed that in general, Macau's spatial structure exhibits a wheel-and-shaft 'core-periphery' feature, and the overall integration of the road network is gradually decreasing from the centre to the periphery layer by layer. The core area of Macau's overall integration includes a contiguous high-integration area bounded by Haibian New Street in the northwest, Meifujiang Road in the east, Desheng Road in the south, and Friendship Square and Amaral Square in the west. This is the core area of the Macau Peninsula with large flows of people and vehicles. The roads in this area are characterised by high integration, accessibility, centrality, and penetrability. At present, many of the gaming industries in Macau are located in this area to reap the benefits of

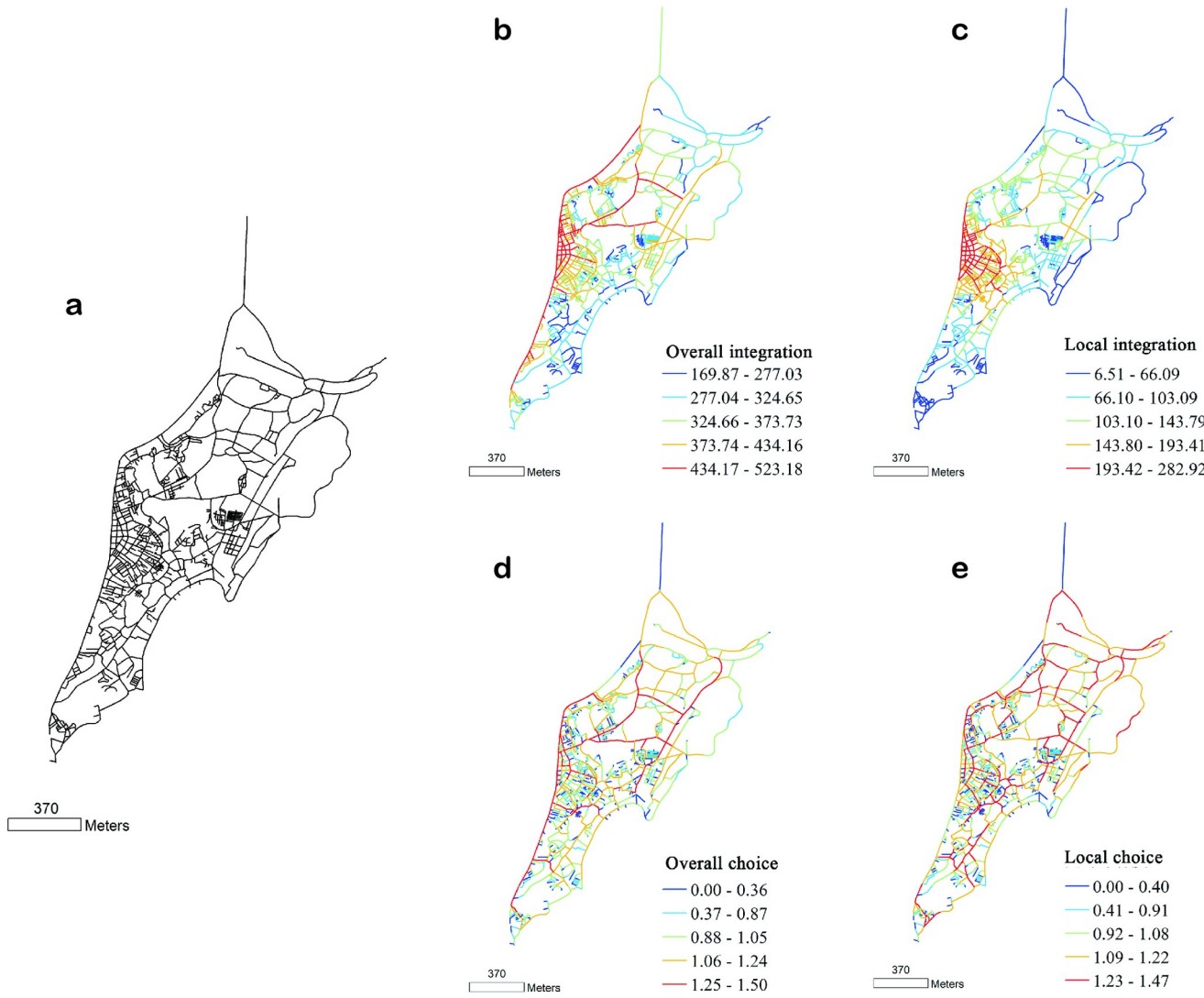

**Fig 3. (a) Structure, (b, c) integration, and (d, e) angle choice of Macau streets and lanes, drawn according to the 1889 base map.** Global variable radius = n; local variable radius = 500 m.

road accessibility. The local accessibility in this area reveals the characteristics of the urban spatial structure with recognition of the local traffic flow (mainly pedestrian flow). Based on its characteristics, the overall choice reflects the passing traffic flow through the road network, and the sections with higher overall choice are often those with higher commuting traffic in the area. The analysis results indicate that the overall choice forms a network structure but does not form a core structure. A few major roads on the Macau Peninsula have a high overall choice.

## Intelligibility of historical maps of streetscape in Macau

The degree of intelligibility can reflect the relationship between the features of the local road scale and the overall road scale in an area. Intelligibility is often characterised in terms of the coefficient of determination of the scatter plots of the local integration and the overall integration and their fitted straight line. Fig 5 presents a numerical analysis of the intelligibility of the

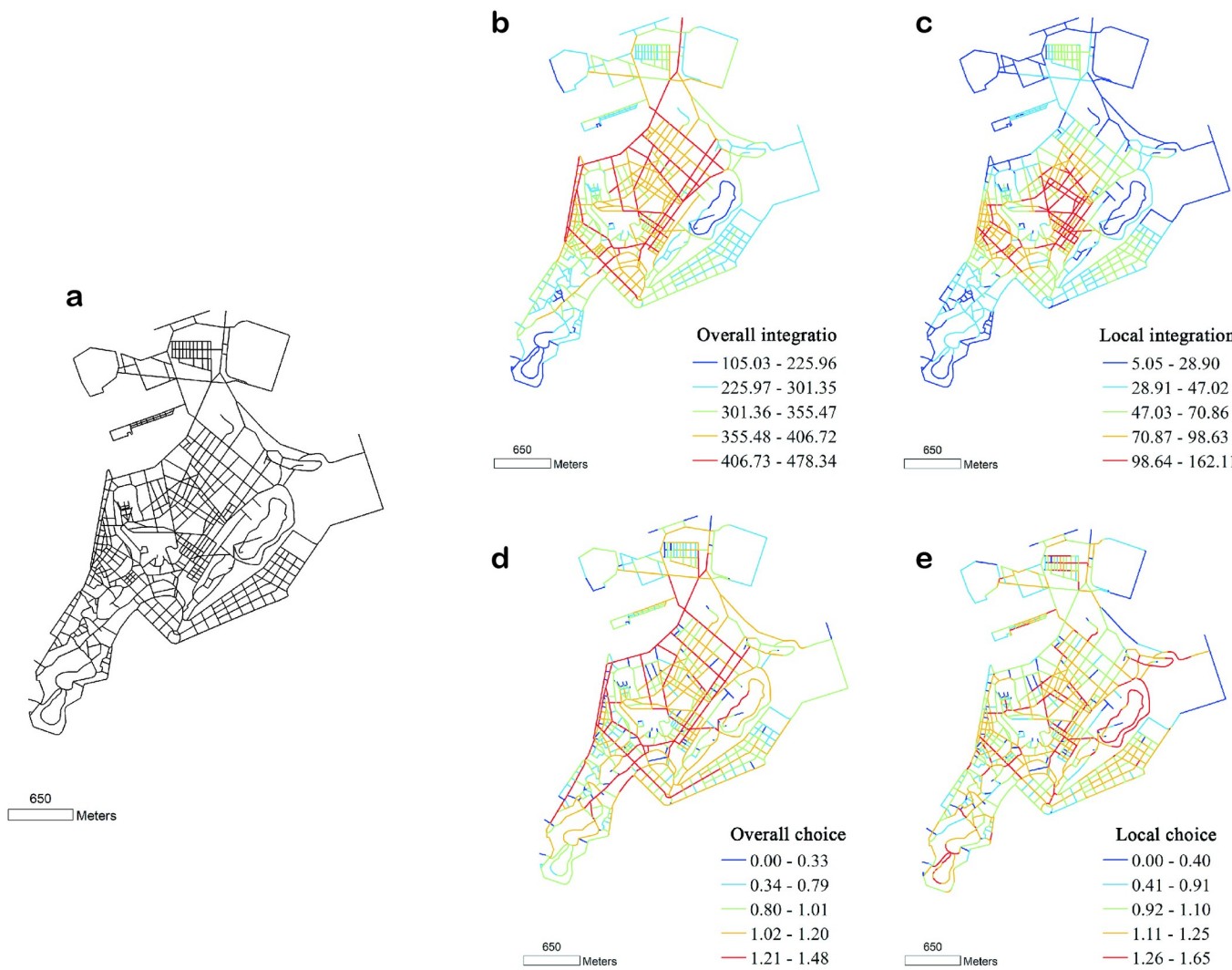

**Fig 4. (a) Structure, (b, c) integration, and (d, e) angle choice of Macau streets and lanes, drawn according to the 1950 base map.** Global variable radius = n; local variable radius = 500 m.

Macau Peninsula in 1780, 1889, 1950 and 2018. Linear regression analysis of global and local integration degrees was performed out using DepthMap software to obtain an understandability scatter diagram (Fig 5). Intelligibility is a variable that describes the correlation between local and overall integration. The x- and y-axis represent the overall integration and R3 local integration, respectively, and $R^2$ is the degree of correlation between the two. The larger is $R^2$, the greater is the degree of correlation. When $R^2 > 0.7$, the degree of correlation between the overall and local integration is highly strong. When $R^2 > 0.5$, this degree of correlation is moderately strong. However, when $R^2 < 0.5$, the scattered points are dispersedly distributed on both the sides of the regression line for the quadratic equation of one variable. If the fitting effect of the trend line is not ideal, the overall and local integration are weakly correlated or uncorrelated. As shown in Fig 5, we used it to analyse the cognitive relationship between the local and overall spaces of the street and lane structure. Here, the x-axis (local integration) represents the global integration degree, the y-axis (overall integration) represents the local integration degree, $R^2$ (Index in the upper right corner of the table) denotes the degree of

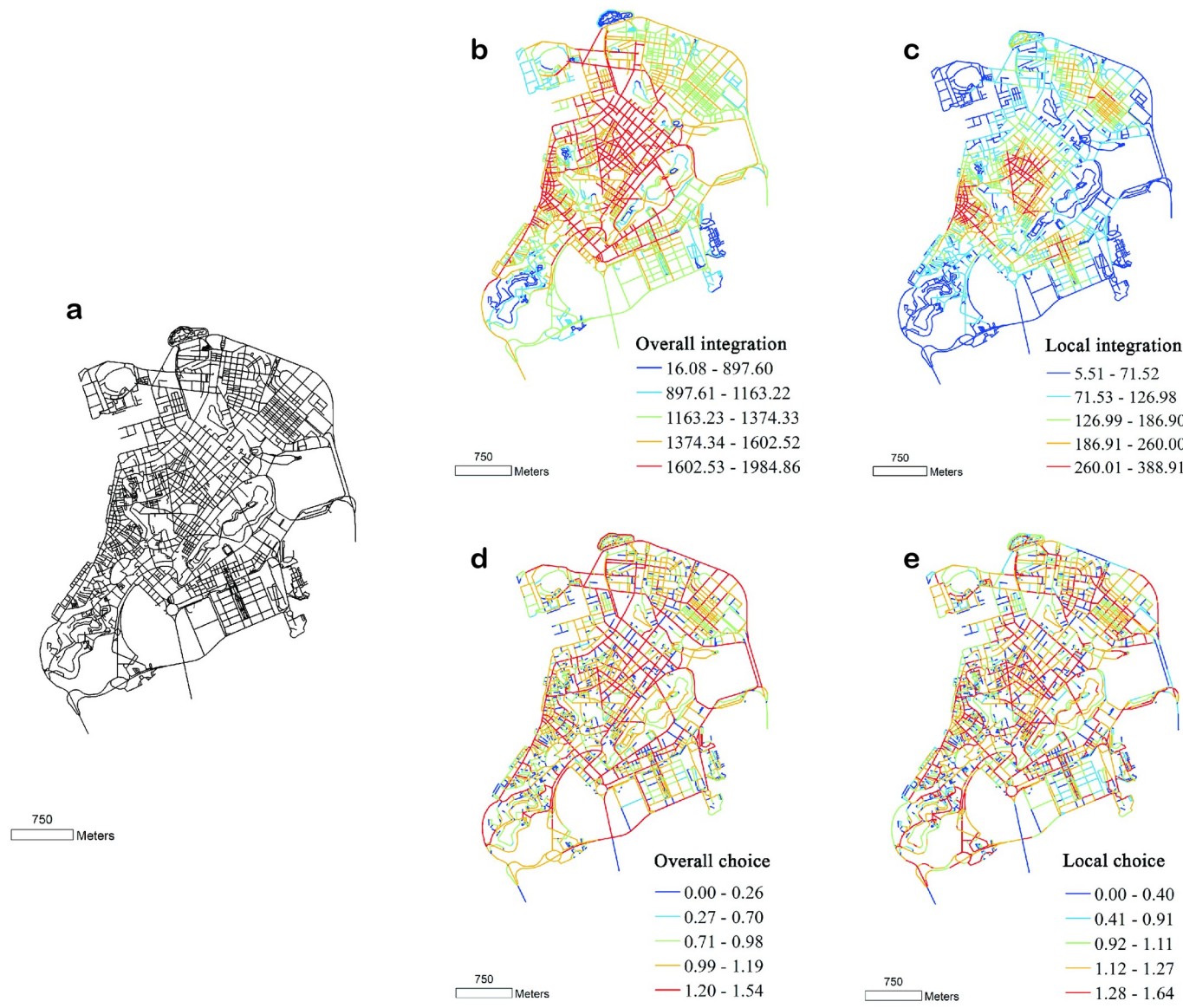

**Fig 5. (a) Structure, (b, c) integration, and (d, e) angle choice of Macau streets and lanes, drawn according to the 2018 base map.** Global variable radius = n; local variable radius = 500 m.

intelligibility, and the fitting coefficient of global integration degree and local integration degree.

As shown in Fig 6, the top right of each plot represents the intelligibility coefficient for the four periods. The results indicate that the degree of intelligibility was the highest in 1780 (0.605). With the passage of time, it exhibited a downward trend, but in 1950, it rebounded slightly, followed by a large decline.

## Discussion

Macau is a unique witness to the first contact and collision between China and the West, which lasted for an extended period. From the 16th to 20th centuries, Macau was a meeting point for merchants, missionaries, and different fields of learning. This meeting of different

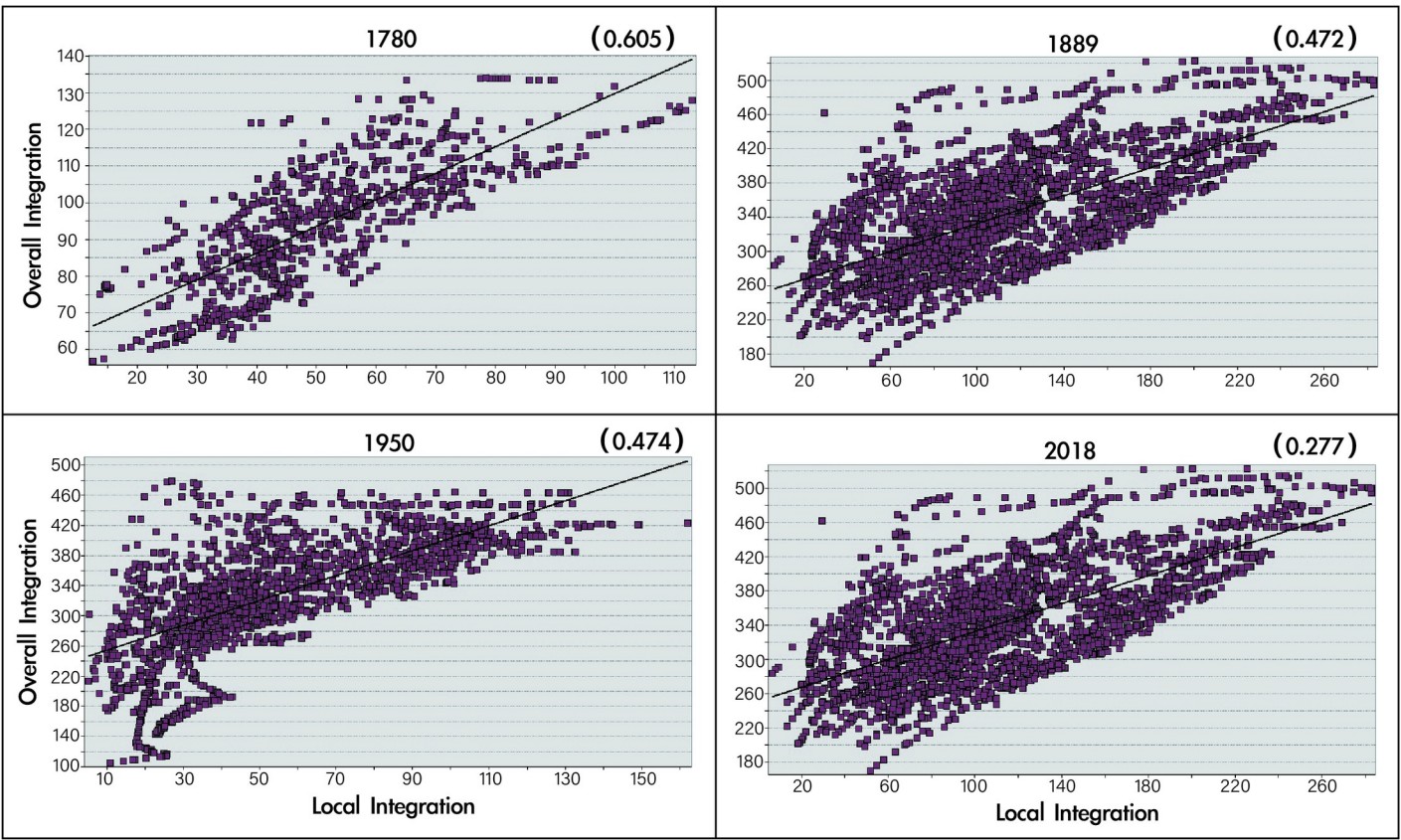

**Fig 6. Numerical analysis of intelligibility of the Macau Peninsula in 1780, 1889, 1950, and 2018.**

cultures can be traced in the region at the core of Macau's history. Macau is an outstanding architectural repository that exemplifies the collision and development of the Chinese and Western civilisations over four and a half centuries. Macau is represented by a series of spaces and architectural groups that connect ancient Chinese ports to Portuguese cities through historical routes. On July 15, 2005, the historic district of Macau met the selection criteria and was included in the World Heritage List, making it the 31st World Heritage Site in China. The street landscape of Macau inherits the characteristics of the integration of Chinese and Western cultures in Macau, which deepened in the process of stratification and emerged as an important feature of the city.

## Structural layering of the streetscape in Macau

In the urban streetscape of Macau, the distribution of streets and lanes is affected by geographical location, social and historical events, and people, exhibiting different street structures and shapes [41]. For example, the streetscape in the waterfront, starting from the waterfront, exhibits a structure that extends and radiates from the waterfront toward the inland, and the residential areas in Macau or places with landmarks and clusters exhibit a radial structure. Such radial structures are often centred around churches or residential areas and radiate towards the surroundings. The reclamation area in Macau is a spatial structure that has gradually expanded over time and is mostly used for commercial purposes [42]. It represents another general streetscape structure, that is, either rectangular or grid-like. There are other regular streetscape structures, and their mechanical characteristics represent later plans (Fig 7). Some

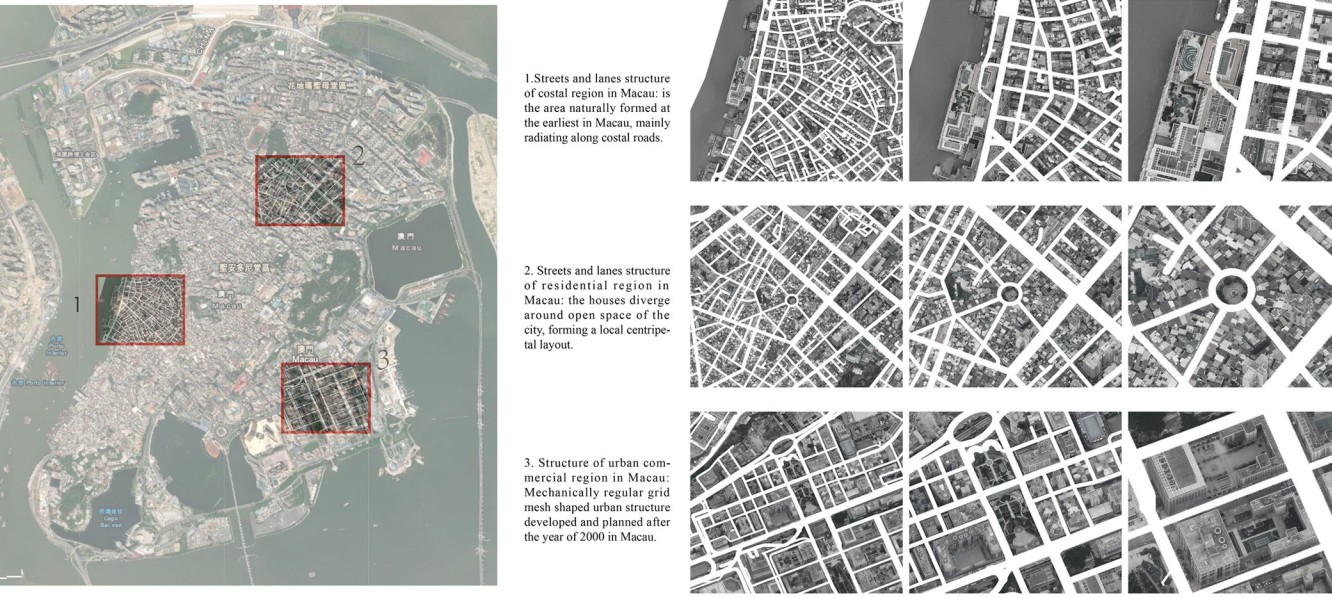

**Fig 7. Structural features of the streetscape in Macau.** Reprinted from [http://earthobservatory.nasa.gov/] under a CC BY license with permission from [NASA Earth Observatory], original copyright [2019].

reasons for the formation of these diverse structures are as follows. First, the road network of the Macau Peninsula is highly compact due to the construction boom of new reclamation areas. Second, restricted by the complex geographical conditions, in the process of rapid urban expansion and construction, the streets pointing from the edges of the original city to the city centre were not built timely. Therefore, in the new districts, the road network structures that were developed based on the original road network show a unique 'collage' mode, without significant changes to the central structure.

The straight street of the city originated in the Iberian Peninsula. It is one of the most important streets in Macao. During Germanic rule, the construction of the straight street started to gradually subvert the old urban spatial distribution of the Romans. The straight street of a city can be regarded as a guiding main road. It is straight or curved and long or short. It dominates the urban spatial distribution design and structure types [43]. The straight street is an extremely important street in the urban space of Portugal. It connects several key nodes, landmarks, and areas in the city. This structure improves accessibility for the city's traffic. Moreover, the city becomes a whole, which increases the convenience of the overall planning and layout design of the city. Examples of urban construction using straight streets are not uncommon, such as Taormina in eastern Sicily, Italy, where a main road connects cities built on two hills, and the famous tourist city of Venice, where a canal with an S-shaped layout connects the economic and political centres of the city [44]. The straight street of Macau has been preserved on historical maps (Fig 8).

## Analysis of explicit characteristic layering rule of the streetscape in Macau

From the land reclamation perspective, from the 1850s to 1970s, the Macau–Portuguese government approved or promoted at least eight reclamation projects, including the Xiahuan Street reclamation, Taipa Tin Hau Temple reclamation, reclamation in front of the A-Ma Temple, reclamation in front of Kanggong Temple, reclamation in Shalitou, and reclamation in Baiyantang. These projects have added large tracts of land area to the western part of the

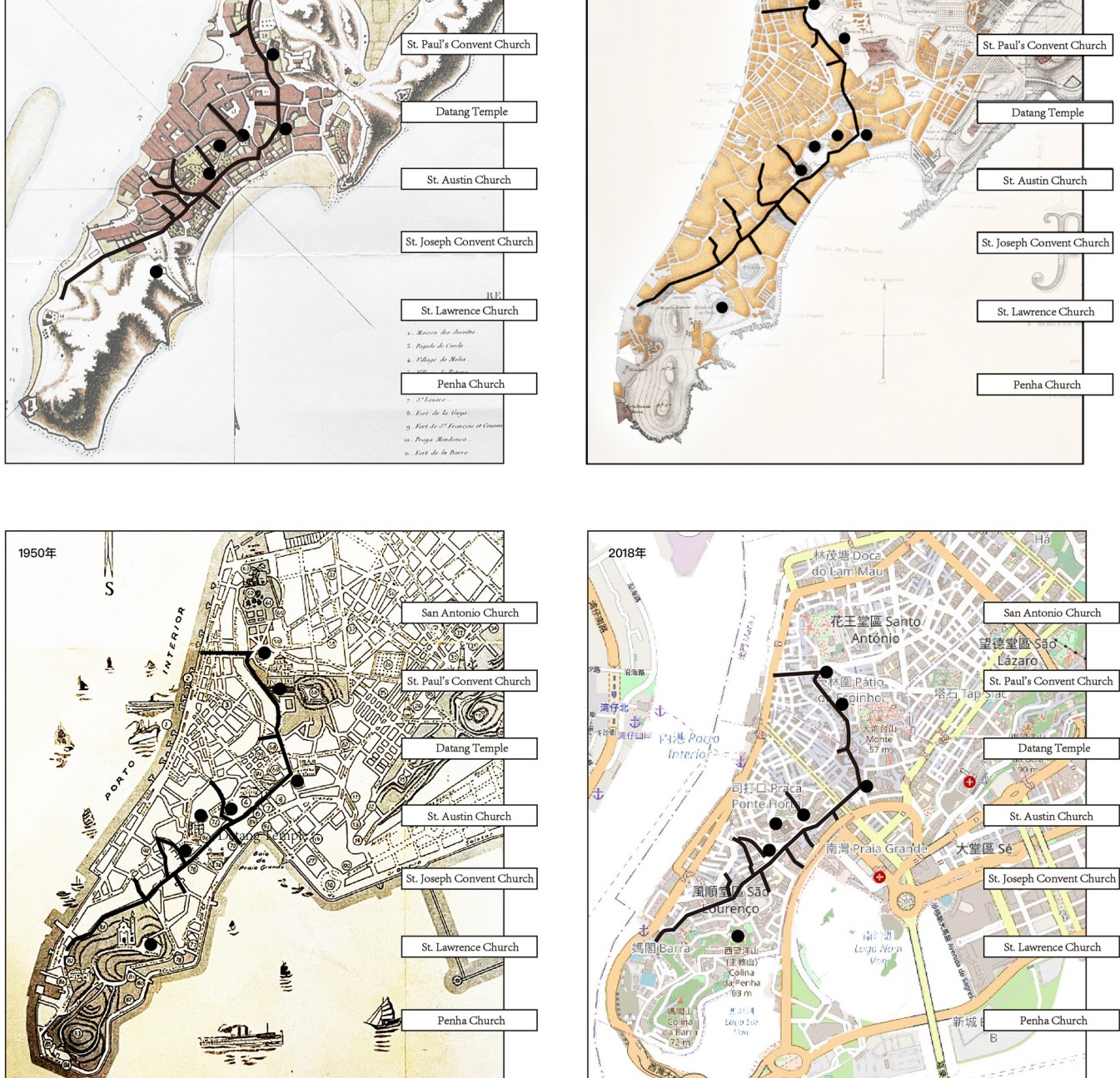

**Fig 8. Historical layering of straight streets in Macau [45]. (1780, 1889, 1950, and 2018).** Reprinted from [MNL.05.03. CART 1780] under a CC BY license with permission from [ARQUIVO DE MACAU], original copyright [1780]. Reprinted from [MNL.10.18h. CART 1889] under a CC BY license with permission from [ARQUIVO DE MACAU], original copyright [1889]. Reprinted from [MNL.05.26. CART 1950] under a CC BY license with permission from [ARQUIVO DE MACAU], original copyright [1950]. Reprinted from [https://viewer.nationalmap.gov/advanced-viewer/] under a CC BY license with permission from [USGS National Map Viewer], original copyright [2018]. Image source: The author selected the historical maps of 1780, 1889, 1950, and 2018 as the base maps for layering analysis and self-drawing. The base maps of 1780, 1889, and 1950 were obtained from the Archives of Macau, Cultural Affairs Bureau, Macau S.A.R. Government. The base map of 2018 was obtained from the USGS National Map Viewer. https://viewer.nationalmap.gov/advanced-viewer/.

Macau Peninsula. Based on planning in the reclamation areas, new streets were opened up. These reclamation projects have affected urban development in multiple ways, including expansion of the urban space of Macau, filling up of the beaches to improve sanitation, replanning and construction of roads on reclaimed lands, and changes to the original city layout [46].

The ways in which land reclamation in Macau has shaped the city's historical streetscape is a meaningful issue in the study of the city's historical landscape. Macau, a city where Chinese and Western cultures blend, has spaces that retain the old remnants of the Chinese culture. In the inner harbour half of the mountain line in the southern peninsula, a rich fishing village culture and the Matsu temple can be traced back to the Ming Dynasty. After Macau became a Portuguese colony, this small corner served as a loading and unloading port for Chinese trade ships. With the development of reclamation engineering technology, a rectangular block was added to this area along the coastline. With the passage of time, the location of the coastline on the map has constantly changed [47]. With the superposition of the newly added land space onto the original space, the coastal streets next to the original coastline of the streetscape gradually became inland streets, and they were superimposed again in the subsequent reclamation. The coastline was moved further out, and the streetscape changed accordingly.

## Analysis of the intelligibility of historical streetscape maps of Macau

The degree of intelligibility was the highest in 1780. At this time, the Portuguese government of the Macau Peninsula had not yet implemented the 'Demolition of the Wall' decision (absolute colonial plan). The Macau Peninsula retained a high-purity aboriginal culture. This cultural gene is rooted in the spatial structure of the roads in the long-term history of the city [48]. Under the influence of this cultural symbol, the historical streets in Macau Peninsula have developed a coherent feature with high intelligibility, which is easily recognisable by the users of the space. In the middle of the 19th century, the Portuguese government implemented the famous 'absolutely autonomous colony' policy. The city wall was demolished, and the Portuguese government completely entered the Macau Peninsula. At this time, the cultural genes originally embedded in the road network were disturbed for the first time. Urban construction under the influence of Portuguese culture gradually transformed the street form of the Macau Peninsula, and new cultural genes were incorporated into the streets of the Macau Peninsula. This type of cultural and paradigm 'invasion' in urban construction leads to two results. One result represents the self-defence and defensive characteristics of traditional historical networks. For example, for open spaces and historical streets, such as the straight street, although the external structure was impacted, the internal features retained the road network form that existed before the implementation of the 'absolutely autonomous colony' plan. The other result represents the integration of the traditional historical streets with the urban construction and culture of the Macau Peninsula under the complete rule of the Portuguese government, leading to the formation of a multicultural street. Both results lead to a decline in the recognition and intelligibility of the road network. Therefore, according to the syntax analysis data, the author examined the characteristics of the urban structure under the impact of such multiculturalism.

After the return of Macau in 1999, the Chinese government and Chinese inland culture had their own impacts on the streets in the Macau Peninsula. At this time, the pace of urban construction in Macau accelerated, and the degree of urbanisation gradually increased. With this background, the intelligibility of Macau Peninsula decreased to 0.277. The current intelligibility of Macau Peninsula cities is low, which is reflected in the poor recognition of Macau's blocks by people such as tourists, and it is easy to 'get lost' in the city. This is because the

intelligibility and cultural diversity of the Macau peninsula are negatively related. Due to multiculturalism and the fact that the city cannot be expanded on a large scale due to terrain limitations, the Macau Peninsula has exhibited low intelligibility.

## Conclusions

The layering map in Fig 8 indicates that the urban streetscape developed through a layered evolution process from the west to east and then from the north to south. Throughout the development of the city, the old Macau city always existed as an important street hub, especially the new road, which houses the important commercial and tourist hubs of the city. As an important part of the city's historical landscape, the new street is home to commerce, culture, and history, in addition to being an important node throughout the historic layering of the city. However, the construction of the streetscape in Macau is closely related to Macau's urban expansion and land reclamation projects, which provided considerable space for the expansion of urban streets [49].

Similar to the research of Griffith *et al.*, from the perspective of urban evolution, space syntax can be concluded as the practice of map and 'mapping'. The space syntax analysis conducted using the quantitative description of urban road network can be used to formulate and test hypotheses about the patterns of urban movement, encounter, and socioeconomic activities in the past that can help interpret other historical source materials to acquire an overall account of the urban spatial culture. In this study, we performed an in-depth quantitative analysis of the street and lane network of Macau in different periods by using space syntax in the GIS environment. We used a scientific method to explore the distinctive features of the historical streetscape in Macau and the value relations of the economic, social, and cultural development of the city at the same time to reveal the close interactive relationship between spatial form and human behaviour. The law of urban evolution, as studied from the perspective of historical layering, reflects the wisdom of predecessors in building cities. It provides a clear context for urban development, and the experience and lessons contained in the law provide valuable reference for the development of modern cities. In the streetscape of Macau, the integration of Chinese and Portuguese cultures is the main line running through the historical layering process of the city. The traces of China–Portuguese cultural integration serve as a testimony to the history of the city while highlighting the special cultural imprint of the times. The changes in the road network of Macau reveals that the urban evolution of the Macau Peninsula is characterised by the construction of new urban areas. The old city mainly involved small-scale reconstructions, thus retaining its original spatial structure. The new urban area, however, focused on the construction of reclamation areas, showing different construction methods and planning layouts. The road network in the Macau Peninsula has experienced constructions in different historical periods, which highlights its unique 'collage' characteristics in the spatial layout. In addition, our study of the initial stage, growth stage, heyday stage, and decline and regeneration stage revealed that even with the extensive development of the city, the old city of Macau has always existed as an important street hub. The old streets in the old city have a unique charm, such as the 'straight street', which is one of the characteristic streets of Macau that must be protected and inherited in the urban development process. In this paper, the relative importance of each space in the entire space system is studied by analysing image data. Through an analysis of the historical urban streetscape of Macau, we discovered that as an important part of the historical urban landscape, the old streets have great commercial and tourist value; represent commerce, culture, and history; and serve as important nodes throughout the historic urban layering.

The historical urban streetscape of Macau presents a rich variety of forms. Some structures are long and short, some are upright, and some are curved, and they are connected to the

characteristic open spaces of Macau. Each building as a node in a street serves as an important memory in the historical urban landscape of Macau. In future studies, we will focus on how to inherit the intrinsic value of the urban historical landscape of Macau. In addition, we will focus on the application of digital technology and the construction of an urban historical landscape digital heritage information protection platform to provide the necessary information support and decision-making tools. Furthermore, based on stratification research, to improve the scientific nature of heritage research, protection and management are other key directions for future research and scholarship.

## Supporting information

**S1 File.**
(DOCX)

## Acknowledgments

The authors appreciate the work of the editor and the reviewers. Thanks to Mr. Hengyu Gu of Peking University for his help with this thesis.

## Author Contributions

**Conceptualization:** Yue Huang.

**Resources:** Yue Huang.

**Software:** Youping Teng, Yue Huang.

**Supervision:** Youping Teng.

**Visualization:** Yue Huang.

**Writing – original draft:** Youping Teng, Shuai Yang, Yue Huang.

**Writing – review & editing:** Shuai Yang.

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
