## [Decision Letter · Decision Letter 0]

8 Jan 2021

PONE-D-20-36948

Research on the Historical Layering of Urban Streetscape in Macau Based on Space Syntax

PLOS ONE

Dear Dr. Yang,

Thank you for submitting your manuscript to PLOS ONE. After careful consideration, we feel that it has merit but does not fully meet PLOS ONE’s publication criteria as it currently stands. Therefore, we invite you to submit a revised version of the manuscript that addresses the points raised during the review process.

We look forward to receiving your revised manuscript.

Kind regards,

Bing Xue, Ph.D.

Academic Editor

PLOS ONE

Journal Requirements:

4. We note that Figures 1, 2, 4, 7, and 10 in your submission contain map images which may be copyrighted. All PLOS content is published under the Creative Commons Attribution License (CC BY 4.0), which means that the manuscript, images, and Supporting Information files will be freely available online, and any third party is permitted to access, download, copy, distribute, and use these materials in any way, even commercially, with proper attribution. For these reasons, we cannot publish previously copyrighted maps or satellite images created using proprietary data, such as Google software (Google Maps, Street View, and Earth). For more information, see our copyright guidelines: http://journals.plos.org/plosone/s/licenses-and-copyright.

(1) You may seek permission from the original copyright holder of Figures 1, 2, 4, 7, and 10 to publish the content specifically under the CC BY 4.0 license. 

5. Please ensure that you refer to Figures 2, 4, 5, 8, 10, and 17-21 in your text as, if accepted, production will need this reference to link the reader to the figure.

Reviewers' comments:

Reviewer's Responses to Questions

**Comments to the Author**

1. Is the manuscript technically sound, and do the data support the conclusions?

Reviewer #1: Partly

Reviewer #2: Partly

2. Has the statistical analysis been performed appropriately and rigorously? 

Reviewer #1: I Don't Know

Reviewer #2: Yes

3. Have the authors made all data underlying the findings in their manuscript fully available?

Reviewer #1: No

Reviewer #2: Yes

4. Is the manuscript presented in an intelligible fashion and written in standard English?

Reviewer #1: No

Reviewer #2: No

5. Review Comments to the Author

Reviewer #1: General Comments

The authors made no attempt to make their case as to why this paper should be published. The abstract and introduction do not explain why the research matters. They do not describe the literature gap or main research question, nor do they put the research in context with other previous work. Both should be rewritten.

In addition to a rewrite of the abstract and intro, the paper would benefit from an intensive edit for structure, flow, and clarity. Specifically, the methods and results sections need to be reordered and restructured. Results need to be rewritten to focus on the main findings, which are confused by having too many figures. Discussion describes findings that were not presented in the results section. Intro and conclusions should mirror each other in structure.

The authors included no line numbers or page numbers, which made the review difficult. I made suggestions for structure that would improve the manuscript; I do not point to specific text that needs changes.

I could not find info about data deposition in the manuscript. Given that the authors produced shapefiles from four different time periods and calculated intelligibility for each time period, I would expect a substantial amount of geospatial data was produced. What are the plans for depositing the data?

Abstract

The abstract does not provide enough detail explaining study. Consider writing five sentences that will set the structure for the rest of the paper.

1: What do we know?

2: What we don't know?

3: How did you fill knowledge gap described in sentence 2?

4: What did you find out?

5: Importance of your work, broader implications, future research

Based on abstract, I don't know why the urban streetscape is important or what its historical value is. Authors say their results deeply explore Macau streetscape but do not describe those results or put them in a broader context.

Introduction

The intro does not provide enough detail to make a compelling case for the study. Why is layering important? Why should you use spatial syntax theory? What are the alternatives? What literature gap are you filling? What are the contributions of your paper?

The introduction is two paragraphs and superficially describes layering. I expected to see more urban morphology or time and space urban literature.

Lynch K. What time is this place?. MIT Press; 1972

Matos Wunderlich FI. Walking and rhythmicity: Sensing urban space. Journal of Urban Design. 2008 Feb 1;13(1):125-39.

Sheng N, Tang UW, Grydehøj A. Urban morphology and urban fragmentation in Macau, China: island city development in the Pearl River Delta megacity region. Island Studies Journal. 2017 Nov 1;12(2):199-212.

Chung T. Valuing heritage in Macau: On contexts and processes of urban conservation. Journal of Current Chinese Affairs. 2009 Mar;38(1):129-60.

Griffiths S, Jones CE, Vaughan L, Haklay M. The persistence of suburban centres in Greater London: combining Conzenian and space syntax approaches. Urban Morphology. 2010;14(2):85-99.

A 5-7 paragraph intro structure to consider:

P1: This is important idea but there are problems with it

P2: Problems/challenges with it are these (x,y,z)

P3: Implications of paragraph 2

P4: Here is how we approach the problem

P5: Contributions of paper and its structure

(you may need up to 2 more paragraphs)

Materials and Methods

This section needs to be reordered. Space syntax theory needs to be in the introduction. Then, the authors need to explain how they collected data, processed it, and analyzed using the specific space syntax methods implemented in the study. Look at Ma et al. 2020 [13] as an example for how you could structure this section in addition to my comments below.

Authors say that intelligibility is a correlation and don't say what kind of correlation (Pearson?). Figures 17-20 are plots of intelligibility (overall integration on y axis and local integration on x axis) and have a regression line. Figure 21 plots intelligibility coefficients for each time period.

I believe that the authors performed regression analysis, and I want the models summarized in a table. Adjusted R2 for model plus coefficients for all variables in models.

I found no description of possible error sources.

The last two paragraphs of the section still contain material from a template (language regarding datasets and animal care). This should be removed.

Consider the following structure:

2.1 Study Area - More detail describing Macau. Authors assumed all readers have familiarity with study area, which will not be true. Include some brief historical context covering your time periods of interest: governance, population, economics, social change, etc.

2.2 Data Acquisition and Processing

Include all details of how you obtained data and processed it. Do not include any analysis here. Something like this:

We obtained official historical maps for the years _____, ____, ___, ___ [data sources]. Using ARCGIS (version?) and CAD (version?), we vectorized each map (how specifically?).

2.3 Data Analysis

Only include data analysis here:

"We imported the calibrated data into Depthmap (source, version?). For each time period, we calculated the following spatial syntax variables (name them all.) Then, discuss how you did the spatial syntax analysis specifically. Explain how you calculated each element, using the formulas for angle integration, angle choice, intelligibility etc. Provide citations for each formula.

Results and Discussion

Results are unfocused, because authors have too many figures. Some of the results are included in the discussion as well. If the authors are going to have separate results and discussion sections, then results need to be concisely presented.

Then, in discussion, interpret the results specifically for Macau context, and then compare to other studies that have applied similar techniques. The authors cite 5 sources in the discussion: 4 relating to Macau and 1 to Naples, Italy.

Discussion brings in some historical context (4 references), but for a reader not familiar with Macau context, it is not easy to link that context with specific results.

The last line of the discussion (below) contains the reference to Naples [26], but it is not clear whether/how/if the Neapolitan context applies.

" Therefore, from the syntactic analysis data, the author examines the characteristics of the urban structure under the impact of such multiculturalism. [26]."

Conclusions

The conclusion should mirror the structure of the intro. Succinctly explain what you did, briefly describe your findings, and their broader implications. What are some future research directions? What are the planning implications of your work?

Figures in General

Too many figures, so the story is lost. I recommend reducing the total number of figures to 6 maximum to best show the patterns of change and to focus the narrative of the paper. Incorporate source of image or analysis into caption.

Consider combining some of your figures so that the trends across each time period are more obvious.

Figures specifically:

1: Move to supp info. Change caption (e.g., Example structural features of streetscape in Macau at three different spatial scales, based on 2019 map (cite Google maps). Then, label scale on top (assuming they are zoomed in to same spatial scale for each box).

2: Keep in main paper. Historical layering of straight streets in Macau, based on maps from 1796, 1889, 1953, and 2010 [data source].

3-4, 7,8, 10,11,13. Delete or move to supplementary info. These maps are source data for your analysis.

5,9,12,14,15 - Keep. These are your analysis results. Interpret these.

15 - Excellent figure. Describe in results; interpret in discussion.

17-21. Can these be combined into a four panel figure for each time period, with the coefficient displayed in the upper right?

References - authors only cite 26 sources. More are needed.

Strengthen introduction and putting your results in context of Macau and other cities where others have applied these methods,

Reviewer #2: 1. The “Abstract” section should include background, methods, results and conclusions. The background should be concise 1-2 sentences, and the results should be the major findings of your study section. Please modify this section accordingly.

2. The current “1. Introduction” section mainly introduces the concept of historical layering. Authors are highly encouraged to provide the relevant background knowledge necessary for the readers to understand why the findings of the paper are an advance on the knowledge in the field. You also need cited more new references, and give the research gap more clearly.

3. The “2. Materials and Methods” section introduced space syntax, but it is not explained why the spatial syntax is used to study the historical layering of urban streetscape in Macau. It is recommended to explain it in the introduction or the methods section.

4. The “4. Discussion" needs to be improved, because it presents only discussion of the results obtained, without correlating with the scientific literature, thus, demonstrate the importance of the results obtained. Besides, the analysis on intelligibility of historical maps of streetscape in Macau should be moved to the results section and not fit into the discussion.

5. In the “5. Conclusions” section, the statement that the names of streets are a major feature of the historical landscape of Macau is a new knowledge, which has little to do with the subject of the article. It is recommended that the main findings be stated in conclusions.

6. PLOS authors have the option to publish the peer review history of their article (what does this mean?). If published, this will include your full peer review and any attached files.

Reviewer #1: No

Reviewer #2: No

---

## [Author Response · Author response to Decision Letter 0]

25 Mar 2021

Reviewer #1: General Comments

The authors made no attempt to make their case as to why this paper should be published. The abstract and introduction do not explain why the research matters. They do not describe the literature gap or main research question, nor do they put the research in context with other previous work. Both should be rewritten.

According to your opinion and the following detailed suggestions, this part has been explained in detail in the Abstract and Introduction. In the revision part, the author added questions such as why this paper should be published, why the research matters, and describe the main research question

The Abstract and Introduction have been rewritten.

In addition to a rewrite of the abstract and intro, the paper would benefit from an intensive edit for structure, flow, and clarity（. Specifically, the methods and results sections need to be reordered and restructured. Results need to be rewritten to focus on the main findings, which are confused by having too many figures. Discussion describes findings that were not presented in the results section. Intro and conclusions should mirror each other in structure. 

I also found such problems in the revision process. And I have made great adjustments and modifications to the structure and content of the article according to the suggestions, which can be shown in the revised article.

The authors included no line numbers or page numbers, which made the review difficult. I made suggestions for structure that would improve the manuscript; I do not point to specific text that needs changes.

Sorry for the trouble caused to the Reviewer. Line Numbers and Page Numbers have been added

I could not find info about data deposition in the manuscript. Given that the authors produced shapefiles from four different time periods and calculated intelligibility for each time period, I would expect a substantial amount of geospatial data was produced. What are the plans for depositing the data?

In this paper, all spatial objects in the spatial syntactic analysis software are modeled based on SSOODM, formed axis map to calculate free spatial morphological variables, and stored on ArcGIS platform. The intelligibility index was obtained after running the syntax software AXWOMAN 6.3. The data of local integration degree and global integration degree are imported into Excel, and the correlation analysis between the determination coefficients of the fitting line is made. Finally, the comprehensibility of the four historical periods is plotted and the analysis results are obtained.

Abstract

The abstract does not provide enough detail explaining study. Consider writing five sentences that will set the structure for the rest of the paper.

1: What do we know?

In 2011, UNESCO pointed out in "Recommendations on Urban Historical Landscapes" that "The historical urban landscape is regarded as the geographical entity of the urban area, and it is understood as the result of the historical accumulation layering f various values."

2: What we don't know?

Although the official put forward the innovative significance and concept of historical layering, there are not too many reference cases when implemented at the operational level. For example, when it is necessary to analyze the complex historical layering, there is less research on how to get rid of theoretical analysis and use quantitative forms to display and analyze.

3: How did you fill knowledge gap described in sentence 2?

The space syntax used in this paper fills the gap of quantitative analysis in historical layering analysis. The article establishes a spatial digital model by combining the urban historical landscape theory with the method analysis of space syntax. A quantitative analysis of the Streetscape in the four historical periods of Macau, and the value-related development of the city's economy, society, and culture. The article uses the theory of urban historical landscape to interpret the Streetscape of Macau.

4: What did you find out?

Urban development is reviewed under different Spatial Scales, presenting different historical layering states. The changing ideological trend of construction brings about the change of the city, which leads to the change of the city style. Its identifiability is constantly changing with the time layering.

5: Importance of your work, broader implications, future research

In the process of analyzing the urban historical landscape, this paper proposes a novel analysis method, which is to use space syntax to quantitatively analyze its spatial form and momentum. The analysis has both the dimensions of space and time, and its results have played an important role in how to continue the life course of the benign growth of the urban landscape and solve the protection problems in practice. At the same time, to provide reference for the protection of historic cities, there is still a lot of work to be done in-depth and unfolded for the future in-depth analysis of more complex historic cities. Through the research of this article, the value of Macau's historical streetscape can be retrieved, and the targeted protection strategy can be implemented so that the historical streetscape of Macau can be passed on.

Based on abstract, I don't know why the urban streetscape is important or what its historical value is. Authors say their results deeply explore Macau streetscape but do not describe those results or put them in a broader context.

As an important part of the regional characteristics of Macao, the urban streetscape of Macao is the continuation of the historical memory of Macao and has irreplaceable historical value. In the context of urban development, the ancient streets always have a unique charm in Macau, such as the “straight streets” of Macau. However, with the advancement of this research, it is found that the unique streetscape of Macau bred by the development of the city does not affect the newly constructed roads. The newly reclaimed area and the streetscape of the new city are on the verge of homogeneity, and basically cannot reflect the unique regional characteristics of Macau. Therefore, Through the research of this article, the value of Macau's historical streetscape can be retrieved, and the targeted protection strategy can be implemented so that the historical streetscape of Macau can be passed on.

Introduction

The intro does not provide enough detail to make a compelling case for the study. Why is layering important? Why should you use spatial syntax theory? What are the alternatives? What literature gap are you filling? What are the contributions of your paper?

The United Nations Educational, Scientific and Cultural Organization (UNESCO) published an instructional collection of essays in 2017 detailing the importance of historical layering for the study of historic urban landscapes，and conducted a detailed case study of historical layering in Cuenca, Ecuador. Macau and Cuenca have many similarities since their initial rise. Both of them used to be colonies, and under the continuous action of time and space, both of them are continuously taking place historical layering process. Urban space or maintain, or replace, or decline, or at the same time, the formation of stratification and juxtaposition as the main situation. The study of UNESCO provides an important theoretical basis for the study of the historical layering of Macao.

Why is layering important?

The law of urban evolution in the study of historical layering contains the great wisdom of predecessors, and provides a clear context for urban development. And the experience and lessons contained in the study provide valuable reference for the development of modern cities. The wisdom of our ancestors, such as cultural tradition, construction technology, morality and etiquette, is worth inheriting and carrying forward. At the same time, it also provides some positive countermeasures for the future development of the city. 

Why should you use spatial syntax theory?

The introduction of space syntax theory is helpful to analyze data quantitatively. The spatial syntax is used to analyze the street and lane network in GIS environment. The very close interaction between spatial form and human behavior is revealed in a scientific way, and the relative importance of each space in the whole spatial system can be easily studied by analyzing data and images. It is a more intuitive and quantitative research method and reflects the form and development of streetscape in this historical period through various syntactic indicators.

What are the alternatives

In the aspect of methodological research on urban historical streetscapes, in addition to space syntax, Kevin. Lynch, Gordon Cullen and other scholars adopted the method of base analysis and made outstanding contributions to the analysis and research of SITE. In his book Site Planning, Lynch put forward a set of Site analysis methods involving social, cultural, psychological, natural and material elements of the context, which has had a significant impact on modern urban design. Cognltive Map is also an effective method to study urban historical streetscapes, which is an urban spatial analysis technology borrowed from the field of cognitive psychology. The concrete process also uses the sociological survey method for reference, which is an effective way to control the urban landscape and place image. The specific approach is to investigate the residents' psychological feelings and impressions of the city through inquiry or written methods, and then let the designers analyze and make them into maps, or more directly encourage them to draw sketches of the spatial structure of the city. In addition, drawing "landscape narrative historical map" is another method, which expresses the changing process of the landscape in a certain region through the presentation form of the map. This method has taken Shanghai as an example to elaborate on the making of its historical and cultural maps.

What literature gap are you filling?

This paper combines the ancient map of Macao with space syntax to explore the quantitative research methods of layering in different historical periods and their applications. 

What are the contributions of your paper?

The concept and method of urban historical landscape give us a new perspective to understand and recognize the layering change of the whole city. The main purpose of this paper is to arouse people's cognition of urban historical and cultural heritage.

The introduction is two paragraphs and superficially describes layering. I expected to see more urban morphology or time and space urban literature.

Lynch K. What time is this place?. MIT Press; 1972

Matos Wunderlich FI. Walking and rhythmicity: Sensing urban space. Journal of Urban Design. 2008 Feb 1;13(1):125-39.

Sheng N, Tang UW, Grydehøj A. Urban morphology and urban fragmentation in Macau, China: island city development in the Pearl River Delta megacity region. Island Studies Journal. 2017 Nov 1;12(2):199-212.

Chung T. Valuing heritage in Macau: On contexts and processes of urban conservation. Journal of Current Chinese Affairs. 2009 Mar;38(1):129-60.

Griffiths S, Jones CE, Vaughan L, Haklay M. The persistence of suburban centres in Greater London: combining Conzenian and space syntax approaches. Urban Morphology. 2010;14(2):85-99.

Thank you very much for the comments of Reviewer. I have read all the above papers carefully and added relevant important contents in my paper, especially the contents of Urban morphology and urban space type of Macao. These are also added in the references.

A 5-7 paragraph intro structure to consider:

P1: This is important idea but there are problems with it

P2: Problems/challenges with it are these (x,y,z)

P3: Implications of paragraph 2

P4: Here is how we approach the problem

P5: Contributions of paper and its structure

(you may need up to 2 more paragraphs)

I have made some modifications according to the suggestion, and the paragraphs of introduction has been extended to 4 

Materials and Methods

This section needs to be reordered. Space syntax theory needs to be in the introduction. Then, the authors need to explain how they collected data, processed it, and analyzed using the specific space syntax methods implemented in the study. Look at Ma et al. 2020 [13] as an example for how you could structure this section in addition to my comments below.

The article is already being reordered. Space syntax theory has been adjusted to the introduction section. For collected data and processed, the author selected the historical maps of 1780, 1889, 1950 and 2018 as the base map for layering analysis and self-drawing. The base maps (1780, 1889, 1950) come from Cultural Affairs Bureau of the Macao S.A.R. Government. The Archives of Macao. The base map (2018) comes from USGS National Map Viewer. https://viewer.nationalmap.gov/advanced-viewer/. First of all, the four times street and lane structure drawings are obtained by using CAD. Then, according to the drawings, the integration degree and crossing degree of the street and lane structure of the four time are derived by using GIS and space syntax, and then the stratification status of the historical streetscape of Macao in different periods is obtained.

Authors say that intelligibility is a correlation and don't say what kind of correlation (Pearson?). Figures 17-20 are plots of intelligibility (overall integration on y axis and local integration on x axis) and have a regression line. Figure 21 plots intelligibility coefficients for each time period.

Correlation is used to describe the degree of correlation between the local control and the overall association. It reflects how easy it is for the subject to distinguish the local space pattern from the overall space pattern. Local space and global space must exist, and there is a certain degree of correlation. The greater the correlation is, the more consistent the local perceptible space is with the overall invisible space, so the spatial form is easier to be understood.

The Numerical analysis of the intelligibility in Figures 17-20 gives a detailed explanation of the process. R2 is explained and the calculation formula is added.

〖R^2=[Σ(C_i-c ® )(I_i-I ® )]/(Σ(C_i-c ® )^2 Σ(I_i-I)^2 )〗^2

I believe that the authors performed regression analysis, and I want the models summarized in a table. Adjusted R2 for model plus coefficients for all variables in models. 

The models have summarized in a table, and I have added the content to the appropriate location as requested.

I found no description of possible error sources.

It has been added in Section 2.3

The last two paragraphs of the section still contain material from a template (language regarding datasets and animal care). This should be removed.

These two paragraphs have been removed.

Consider the following structure:

2.1 Study Area - More detail describing Macau. Authors assumed all readers have familiarity with study area, which will not be true. Include some brief historical context covering your time periods of interest: governance, population, economics, social change, etc.

The Space Syntax section has been moved up to Introduction. From the perspective of society, culture, architecture and so on, more detail about Macau is added. It emphasizes that Macao used to be a colonialist country, and its multi-coexistence and mutual integration of social culture is an important social feature.

2.2 Data Acquisition and Processing

Include all details of how you obtained data and processed it. Do not include any analysis here. Something like this:

We obtained official historical maps for the years _____, ____, ___, ___ [data sources]. Using ARCGIS (version?) and CAD (version?), we vectorized each map (how specifically?).

It has been carefully modified according to the requirements.

2.3 Data Analysis

Only include data analysis here:

"We imported the calibrated data into Depthmap (source, version?). For each time period, we calculated the following spatial syntax variables (name them all.) Then, discuss how you did the spatial syntax analysis specifically. Explain how you calculated each element, using the formulas for angle integration, angle choice, intelligibility etc. Provide citations for each formula. “

This part of the content proposed by the Author has been introduced in detail in the original 2.1 section on space syntax, which has been slightly modified into Section 2.3. The Depthmap version was depthmapX-0.7.0. Adding and refining the formula for the calculation, and Providing citations for each formula.

Results and Discussion

Results are unfocused, because authors have too many figures. Some of the results are included in the discussion as well. If the authors are going to have separate results and discussion sections, then results need to be concisely presented.

Results and Discussion have been merged into one chapter. At the same time, I have revised about the content and structure to make my results look focused. 

Then, in discussion, interpret the results specifically for Macau context, and then compare to other studies that have applied similar techniques. The authors cite 5 sources in the discussion: 4 relating to Macau and 1 to Naples, Italy.

Discussion brings in some historical context (4 references), but for a reader not familiar with Macau context, it is not easy to link that context with specific results.

 “Macao is the unique witness of the first contact and collision between China and the West, which lasted for a long time. From the 16th to the 20th centuries, Macao was a meeting point for merchants, missionaries, and different fields of learning. This meeting of different cultures can be traced in the region represented by the core of Macao's history. Macau is an outstanding representative of the architectural community, which exemplifies the collision and development of Chinese and Western civilization in four and a half centuries. Macau is represented by a series of Spaces and architectural groups that connect ancient Chinese ports to Portuguese cities in the historical route. On July 15, 2005, the historic district of Macao met the selection criteria and was inscribed on the World Heritage List, making it the 31st World Heritage Site in China. The street landscape of Macao inherits the characteristics of the integration of Chinese and western cultures in Macao, which is deepened in the process of stratification and becomes an important feature of the city.” The content of Macao has been added in Discussion.

The last line of the discussion (below) contains the reference to Naples [26], but it is not clear whether/how/if the Neapolitan context applies.

" Therefore, from the syntactic analysis data, the author examines the characteristics of the urban structure under the impact of such multiculturalism. [26]."

There is an error here and the reference has been deleted

Conclusions

The conclusion should mirror the structure of the intro. Succinctly explain what you did, briefly describe your findings, and their broader implications. What are some future research directions? What are the planning implications of your work?

It has been modified according to the requirements

Figures in General

Too many figures, so the story is lost. I recommend reducing the total number of figures to 6 maximum to best show the patterns of change and to focus the narrative of the paper. Incorporate source of image or analysis into caption.

This article has reduced a lot of unimportant images. It has been noted in the revised article. However, a few pictures were very important, so they were not deleted. There were 1-2 more pictures than the review expected.

Consider combining some of your figures so that the trends across each time period are more obvious. Strengthen introduction and putting your results in context of Macau and other cities where others have applied these methods,

Thanks for the comments, I have rewrote the introduction and results.

Figures specifically:

1: Move to supp info. Change caption (e.g., Example structural features of streetscape in Macau at three different spatial scales, based on 2019 map (cite Google maps). Then, label scale on top (assuming they are zoomed in to same spatial scale for each box).

Caption is modified to “Structural Features of Streetscape in Macau at Four Different Spatial Scales, Based on maps (1780\\1889\\1950\\ 2018).”

2: Keep in main paper. Historical layering of straight streets in Macau, based on maps from 1796, 1889, 1953, and 2010 [data source].

It has been kept in main paper. However, considering that some pictures can’t be authorized, we choose pictures (can be authorized) that have a similar year instead.

3-4, 7,8, 10,11,13. Delete or move to supplementary info. These maps are source data for your analysis.

The images have since been deleted

5,9,12,14,15 - Keep. These are your analysis results. Interpret these.

Fig. 5,9,12,14,15 are kept. As Fig.5, 8, 11 and 13 are all the structural drawings of streets and alleys in Macao (in different historical periods), for the article integrity, Fig.5 and 6 are combined into one picture, Fig.8 and 9 are combined into one picture, Fig.11 and 12 are combined into one picture, and Fig.13 and 14 are combined into one picture.

15 - Excellent figure. Describe in results; interpret in discussion.

The location of Figure 15 has been changed. And made a supplement according to the request.

17-21. Can these be combined into a four panel figure for each time period, with the coefficient displayed in the upper right?

It has been modified as required

References - authors only cite 26 sources. More are needed.

On the basis of the original, added more References, there are nearly 50 References

Reviewer #2: 1. The “Abstract” section should include background, methods, results and conclusions. The background should be concise 1-2sentences, and the results should be the major findings of your study section. Please modify this section accordingly.

Regarding the Abstract part, Reviewer 1 also puts forward corresponding suggestions. Based on the opinions of two Reviewers, the Abstract has been rewritten.

2. The current “1. Introduction” section mainly introduces the concept of historical layering. Authors are highly encouraged to provide the relevant background knowledge necessary for the readers to understand why the findings of the paper are an advance on the knowledge in the field. You also need cited more new references, and give the research gap more clearly.

The "Introduction" section has been rewritten according to the requirements, and more references have been added to expand a large part of the knowledge. The "Introduction" section is not only about historical layering, but also about urban historical landscape, space syntax, domestic and foreign research status of this part, etc., to highlight the findings of the paper are an advance on the knowledge in the field.

3. The “2. Materials and Methods” section introduced space syntax, but it is not explained why the spatial syntax is used to study the historical layering of urban streetscape in Macau. It is recommended to explain it in the introduction or the methods section.

About "Why the spatial syntax is used to study the historical layering of urban streetscape in Macau." I have adjusted the structure of the article, explicitly and in detail added this content in "Introduction", and also put forward other alternative methods.

4. The “4. Discussion" needs to be improved, because it presents only discussion of the results obtained, without correlating with the scientific literature, thus, demonstrate the importance of the results obtained. Besides, the analysis on intelligibility of historical maps of streetscape in Macau should be moved to the results section and not fit into the discussion.

The structure of the article has been modified and relevant content added as suggested by Reviewer #2.

5. In the “5. Conclusions” section, the statement that the names of streets are a major feature of the historical landscape of Macau is a new knowledge, which has little to do with the subject of the article. It is recommended that the main findings be stated in conclusions.

The "The Names of Streets" section has been removed as requested. At the same time, the conclusions are revised and the main findings are added.

---

## [Decision Letter · Decision Letter 1]

20 Apr 2021

PONE-D-20-36948R1

Structural Features of Streetscape in Macau at Four Different Spatial Scales, Based on Historical Maps

PLOS ONE

Dear Dr. Yang,

Thank you for submitting your manuscript to PLOS ONE. After careful consideration, we feel that it has merit but does not fully meet PLOS ONE’s publication criteria as it currently stands. Therefore, we invite you to submit a revised version of the manuscript that addresses the points raised during the review process.

We look forward to receiving your revised manuscript.

Kind regards,

Bing Xue, Ph.D.

Academic Editor

PLOS ONE

Reviewers' comments:

Reviewer's Responses to Questions

**Comments to the Author**

1. If the authors have adequately addressed your comments raised in a previous round of review and you feel that this manuscript is now acceptable for publication, you may indicate that here to bypass the “Comments to the Author” section, enter your conflict of interest statement in the “Confidential to Editor” section, and submit your "Accept" recommendation.

Reviewer #1: (No Response)

2. Is the manuscript technically sound, and do the data support the conclusions?

Reviewer #1: Partly

3. Has the statistical analysis been performed appropriately and rigorously? 

Reviewer #1: Yes

4. Have the authors made all data underlying the findings in their manuscript fully available?

Reviewer #1: No

5. Is the manuscript presented in an intelligible fashion and written in standard English?

Reviewer #1: No

6. Review Comments to the Author

Reviewer #1: General Comments

The authors have incorporated reviewer suggestions that have improved the manuscript.

They need to do more work to develop the story of place and space syntax change. Authors should consider looking at other space syntax papers to see how they structured their argument, results, and discussion. Discussion needs to be broadened to relate work to previous literature.

More clarity and precision are needed in the results. I believe the results are not yet in the correct order. I would start writing the results with the current Figure 6, which is a truly beautiful rendering of the evolution of the streetscape through time. Authors use lines 516-523 to say that they analyzed the streetscape, but they never describe what the reader should pay attention to in this figure in results, nor do they cite it until the discussion. What are the important points you would like to describe? Clearly, the network expands in area, and it covers also develops more intersections. Are there any particular locations we should pay attention to?

The manuscript could benefit from further editing to improve paper structure, flow, and grammar. I recommend Stephen Heard’s book “The scientists guide to writing” for guidance on developing and presenting the paper narrative (no, I am not the author of this book nor do I have any affiliation with the author).

Please provide specific info where your data may be found: DOI link, data depository, supplementary data files.

####################

Consider using letters for panels in multipart figures. See Figure 1 example. Figures with panels (1-6, 7,8) are a big improvement.

See Claus Wilke for improving multi-panel figure by thinking carefully about panel labels and axes. https://clauswilke.com/dataviz/multi-panel-figures.html. Look at figure 21.1 and 21.2 in his examples to see how not all panels have axes because adjacent panel axis is same.

Sources and licenses for images should be cited by reference.

Look at a recent PLOS article to get ideas for how to properly format your figures and captions. Note that caption title should be bold and rest of caption is plain text. Note left flush, ragged right, not centered.

###################

Specific Text Comments

Lines 16-52. Abstract needs to be simplified to summarize work and explain its importance. Abstract is more than 300 words and needs to be shortened.

Lines 237-239: convert map sources to numbered references and properly cite them in references.

Line 239: New paragraph beginning with “We vectorized…”

Line 239-242: Run on sentence beginning with “We vectorized...”

Line 242-243: Delete from “In order……models results.

Line 245: Change to “Secondly, we imported…”

Line 247: Change second to last sentence. “Third, we converted calculated result into a shapefile, which we then imported into ArcGIS for mapping and spatial analysis.”

Lines 248-250: Delete last sentence of paragraph

Lines 252-253: Just say why you chose syntax line segment model

Lines 261-264: Move to data acquisition and processing paragraph, line 239.

Line 317: change 1st sentence and provide name of coefficient. We defined intelligibility as the _______ correlation/linear regression coefficient between overall and local …” Name the coefficient you used here and in figure 5. You don’t need to give the formula. You said you have a table in response to comments, but I don’t see a table anywhere.

Line 327: Delete “and Discussion”

Lines 328-331: Delete

Lines 332:340: Move to methods

Lines 341:358 – Cite reference earlier in paragraph or else provide another reference to support the information

Line 361: Cite figure 1 here

Line 379: Label all parts of multi-panel figure with letters and refer to them in caption. Use same basic format for all. Example below.

Fig. 1. Structure (a), integration (b,c) and angle choice (d,e) of Macau streets and lanes, drawn according to the 1780 base map. Global variable radius = n; local variable radius = 500m.

Figure 5: Add period after last year (2018). Remove parentheses and bolding of caption part that says “(Upper right….). Name the correlation coefficient used to calculate intelligibility. Make axis numbers and labels larger. You don’t need to label all axes on all figures if the axis is the same as an adjoining. Y axis labels of 1780 and 1950 are needed, but Y axis of 1889 and 2018 are not. Similarly, omit x axis labels of 1780 and 1889.

Line 525: Change caption to more accurately describe figure. Example below. You do not need to say that the image source is the author’s work.

Figure 6. Road layering of the Macau Peninsula, based on drawings from maps of four different time periods.

Lines 586-595: This section should be in results, before your street syntax analysis. This section provides the needed context for all of your other analyses.

Line 620-622: Begin paragraph with “The degree of intelligibility was the highest in 1780.” Delete everything before this

7. PLOS authors have the option to publish the peer review history of their article (what does this mean?). If published, this will include your full peer review and any attached files.

Reviewer #1: No

---

## [Author Response · Author response to Decision Letter 1]

28 May 2021

Dear Dr. Bing Xue,

Thank you very much for your decision letter and advice on our manuscript (Manuscript # PONE-D-20-36948R1) entitled “Structural Features of Streetscape in Macau at Four Different Spatial Scales, Based on Historical Maps”. We also thank the reviewers for the constructive and positive comments and suggestions. Accordingly, we have revised the manuscript. All amendments are highlighted with track changes in the revised manuscript. In addition, point-by-point responses to the comments are listed below this letter.

We hope that the revision is acceptable for the publication in your journal.

Look forward to hearing from you soon. 

With best wishes,

Yours sincerely,

ShuaiYang 

First of all, we would like to express our sincere gratitude to the reviewers for their constructive and positive comments.

Replies to Reviewer 1

Specific Comments

1. The authors have incorporated reviewer suggestions that have improved the manuscript.

They need to do more work to develop the story of place and space syntax change. Authors should consider looking at other space syntax papers to see how they structured their argument, results, and discussion. Discussion needs to be broadened to relate work to previous literature.

Response: As per the suggestions of the reviewer, we have read a large number of articles about space syntax again and gained a deeper understanding of it. We have added three relevant references, and revised the unreasonable discussions in the previous manuscript, particularly the conclusion.

2. More clarity and precision are needed in the results. I believe the results are not yet in the correct order. I would start writing the results with the current Figure 6, which is a truly beautiful rendering of the evolution of the streetscape through time. Authors use lines 516-523 to say that they analyzed the streetscape, but they never describe what the reader should pay attention to in this figure in results, nor do they cite it until the discussion. What are the important points you would like to describe? Clearly, the network expands in area, and it covers also develops more intersections. Are there any particular locations we should pay attention to?

Response: We have moved Fig. 6 to be the first figure in the results. Also, according to the reviewer’s suggestions, we have analysed the road network of the Macau Peninsula more carefully after the previous line 523, and included this revision in the conclusion.

3. The manuscript could benefit from further editing to improve paper structure, flow, and grammar. I recommend Stephen Heard’s book “The scientists guide to writing” for guidance on developing and presenting the paper narrative (no, I am not the author of this book nor do I have any affiliation with the author).

Response: Thanks for your thoughtful suggestion. This manuscript has been edited and proofread by Sci-Edit Publications.

4. Please provide specific info where your data may be found: DOI link, data depository, supplementary data files.

Response: The Macau streetscape maps in the four periods are based on the historical maps. Specifically, the historical maps were first vectorized and had their line segment models created. Then, they were drawn manually and fed into the Depth Map software for data processing.

Title：Structural Features of Streetscape in Macau at Four Different Spatial Scales, Based on Historical Maps，Information files

Link：https://figshare.com/s/b21949bb519df293fe51

5. Consider using letters for panels in multipart figures. See Figure 1 example. Figures with panels (1-6, 7,8) are a big improvement.

Response: Figures 1, 2, 3, and 4 (currently as Figures 2, 3, 4, and 5) have been modified accordingly.

6. See Claus Wilke for improving multi-panel figure by thinking carefully about panel labels and axes. https://clauswilke.com/dataviz/multi-panel-figures.html. Look at figure 21.1 and 21.2 in his examples to see how not all panels have axes because adjacent panel axis is same.

Response: Figures 1, 2, 3, and 4 (Figures 2, 3, 4, and 5 in the current version) have been modified accordingly.

7. Sources and licenses for images should be cited by reference.

Response: The images were drawn manually after vectorizing the historical maps in Fig. 8.

8. Look at a recent PLOS article to get ideas for how to properly format your figures and captions. Note that caption title should be bold and rest of caption is plain text. Note left flush, ragged right, not centered.

Response: Correction has been made in the revised manuscript.

Specific Text Comments

1. Lines 16-52. Abstract needs to be simplified to summarize work and explain its importance. Abstract is more than 300 words and needs to be shortened.

Response: Correction has been made in the abstract of revised manuscript.

2. Lines 237-239: convert map sources to numbered references and properly cite them in references. 

Response: Thanks for your comments. Corrections have been made in the revised manuscript.

3.Line 239: New paragraph beginning with “We vectorized…”

Response: Thanks for your comments. Corrections have been made in the revised manuscript.

4. Line 239-242: Run on sentence beginning with “We vectorized...”

Response: Corrections have been made in the revised manuscript.

5. Line 242-243: Delete from “In order……models results.

Response: Correction has been made in the revised manuscript.

6. Line 245: Change to “Secondly, we imported…”

Response: Correction has been made in the revised manuscript.

7. Line 247: Change second to last sentence. “Third, we converted calculated result into a shapefile, which we then imported into ArcGIS for mapping and spatial analysis.”

Response: Thanks for raising this critical issue. Correction has been made in the revised manuscript.

8. Lines 248-250: Delete last sentence of paragraph

Response: Thanks for raising this critical issue. Correction has been made in the revised manuscript.

9. Lines 252-253: Just say why you chose syntax line segment model

Response: Thanks for raising this critical issue. Correction has been made in the revised manuscript.

10. Lines 261-264: Move to data acquisition and processing paragraph, line 239.

Response: Thanks for suggestion. Correction has been made in the revised manuscript.

11. Line 317: change 1st sentence and provide name of coefficient. We defined intelligibility as the _______ correlation/linear regression coefficient between overall and local …” Name the coefficient you used here and in figure 5. You don’t need to give the formula. You said you have a table in response to comments, but I don’t see a table anywhere.

Response: According to the reviewer’s suggestion, line 317 has been revised, and the equation of R2 has been removed. The coefficient has been named in the figure and the statement about the table has been deleted.

12. Line 327: Delete “and Discussion”

Response: Correction has been made in the revised manuscript.

13. Lines 328-331: Delete

Response: Correction has been made in the revised manuscript.

14.Lines 332:340: Move to methods

Response: Correction has been made in the revised manuscript.

15. Lines 341:358 – Cite reference earlier in paragraph or else provide another reference to support the information

Response: The relevant references have been added.

16. Line 361: Cite figure 1 here

Response: Correction has been made in the revised manuscript.

17. Line 379: Label all parts of multi-panel figure with letters and refer to them in caption. Use same basic format for all. Example below.

Response: Figures 1, 2, 3, and 4 (currently as Figures 2, 3, 4, and 5) have been modified accordingly.

18. Fig. 1. Structure (a), integration (b,c) and angle choice (d,e) of Macau streets and lanes, drawn according to the 1780 base map. Global variable radius = n; local variable radius = 500m.

Response: Figures 1, 2, 3, and 4 (currently as Figures 2, 3, 4, and 5) have been modified accordingly.

19. Figure 5: Add period after last year (2018). Remove parentheses and bolding of caption part that says “(Upper right….). Name the correlation coefficient used to calculate intelligibility. Make axis numbers and labels larger. You don’t need to label all axes on all figures if the axis is the same as an adjoining. Y axis labels of 1780 and 1950 are needed, but Y axis of 1889 and 2018 are not. Similarly, omit x axis labels of 1780 and 1889.

Response: Figure 5 has been modified according to the suggestion.

20. Line 525: Change caption to more accurately describe figure. Example below. You do not need to say that the image source is the author’s work.

Response: Correction has been made here and the statement about the image source has been deleted.

21. Figure 6. Road layering of the Macau Peninsula, based on drawings from maps of four different time periods.

Response: We sincerely appreciate this suggestion and have revised accordingly.

22. Lines 586-595: This section should be in results, before your street syntax analysis. This section provides the needed context for all of your other analyses.

Response: The order has been adjusted and the section in question has been moved to the first paragraph of the conclusion.

23. Line 620-622: Begin paragraph with “The degree of intelligibility was the highest in 1780.” Delete everything before this

Response: We sincerely appreciate this suggestion and have revised accordingly. 

References:

1. Chen Z. The Integration of Chinese and Western Cultures from the Urban Architecture of Macau. M.Sc. thesis, Xiamen University.2002. Available from: https://kns.cnki.net/KCMS/detail/detail.aspx?dbname=CMFD9904&filename=2003042420.nh

2. Battistin Fabiana. Space Syntax and buried cities: The case of the Roman town of Falerii Novi (Italy)[J]. Journal of Archaeological Science: Reports,2021,35.

3. Griffiths, Sam, and Laura Vaughan. "Mapping spatial cultures: contributions of space syntax to research in the urban history of the nineteenth-century city." Urban History 47.3 (2020): 488-511.

4. Zhou L. (2020). A Study on the Evolution of Urban Internal Form in Hangzhou Based on GIS and Space Syntactic (Master's Dissertation, Zhejiang University).

5. USGS National Map Viewer. Open Street Map Macau. 25 Mar 2021.

Available: https://viewer.nationalmap.gov/advanced-viewer

6. 6. Xiaoyu (trans.) (1995).The Chronicle of Aokou (originally by Silva.Beatriz Basto da Silva).Macau: Macau Foundation.

7. 7. Huang Hanqiang, Wu Zhiliang (1996).Overview of Macau.Macau: Macau Foundation.

---

## [Decision Letter · Decision Letter 2]

21 Sep 2021

Structural Features of the Streetscape of Macau across Four Different Spatial Scales Based on Historical Maps

PONE-D-20-36948R2

Dear Dr. Yang,

We’re pleased to inform you that your manuscript has been judged scientifically suitable for publication and will be formally accepted for publication once it meets all outstanding technical requirements.

Kind regards,

Bing Xue, Ph.D.

Academic Editor

PLOS ONE

Additional Editor Comments (optional):

Reviewers' comments:

Reviewer's Responses to Questions

**Comments to the Author**

1. If the authors have adequately addressed your comments raised in a previous round of review and you feel that this manuscript is now acceptable for publication, you may indicate that here to bypass the “Comments to the Author” section, enter your conflict of interest statement in the “Confidential to Editor” section, and submit your "Accept" recommendation.

Reviewer #2: All comments have been addressed

2. Is the manuscript technically sound, and do the data support the conclusions?

Reviewer #2: Yes

3. Has the statistical analysis been performed appropriately and rigorously? 

Reviewer #2: Yes

4. Have the authors made all data underlying the findings in their manuscript fully available?

Reviewer #2: Yes

5. Is the manuscript presented in an intelligible fashion and written in standard English?

Reviewer #2: Yes

6. Review Comments to the Author

Reviewer #2: After revision, the research significance of the article have been strengthened, and the structure has become more reasonable. A detailed description has been added to the method part, and the conclusion part is more complete. I think the paper can be published on PLOS ONE.

7. PLOS authors have the option to publish the peer review history of their article (what does this mean?). If published, this will include your full peer review and any attached files.

Reviewer #2: No

---

## [Editor Report · Acceptance letter]

4 Oct 2021

PONE-D-20-36948R2 

Structural Features of the Streetscape of Macau across Four Different Spatial Scales Based on Historical Maps 

Dear Dr. Yang:

I'm pleased to inform you that your manuscript has been deemed suitable for publication in PLOS ONE. Congratulations! Your manuscript is now with our production department. 

Kind regards, 

on behalf of

Professor Bing Xue 

Academic Editor

PLOS ONE